# Spin-defect qubits in two-dimensional transition metal dichalcogenides operating at telecom wavelengths

Yeonghun Lee ®[1,2] ✉, Yaoqiao Hu[1], Xiuyao Lang[1], Dongwook Kim[1], Kejun Li[3], Yuan Ping ®[4], Kai-Mei C. Fu ®[5,6] & Kyeongjae Cho ®[1] ✉

Solid state quantum defects are promising candidates for scalable quantum information systems which can be seamlessly integrated with the conventional semiconductor electronic devices within the 3D monolithically integrated hybrid classical-quantum devices. Diamond nitrogen-vacancy (NV) center defects are the representative examples, but the controlled positioning of an NV center within bulk diamond is an outstanding challenge. Furthermore, quantum defect properties may not be easily tuned for bulk crystalline quantum defects. In comparison, 2D semiconductors, such as transition metal dichalcogenides (TMDs), are promising solid platform to host a quantum defect with tunable properties and a possibility of position control. Here, we computationally discover a promising defect family for spin qubit realization in 2D TMDs. The defects consist of transition metal atoms substituted at chalcogen sites with desirable spin-triplet ground state, zero-field splitting in the tens of GHz, and strong zero-phonon coupling to optical transitions in the highly desirable telecom band.

An isolated point defect in a crystalline solid can be regarded as an artificial atom whose properties stem from the host material and bonding environment[1–3]. The experimental demonstration of defects exhibiting long spin coherence times ($T_2$) and spin-selective optical transitions have made crystalline point defects one of the most promising platforms for the realization of long-distance quantum networks[1,4]. However, finding a single-point defect that exhibits all the desirable traits for quantum entanglement network generation remains elusive. For example, the popular nitrogen-vacancy (NV) center ($N_C V_C^{-1}$ defect complex in diamond) does not operate at telecom wavelengths for low-loss fiber transmission (optical fiber telecom band: $\lambda = 1260–1675$ nm, or $h\nu = 0.74–0.98$ eV). On the other hand, Er-based qubits do but exhibit small optical oscillator strengths[5,6]. All of the most promising point defects occur in three-dimensional (3D) bulk crystalline materials (diamond[7–9], SiC[10], and oxides[6]), in which

heterostructure fabrication, doping, and device fabrication remain challenging.

Here, we report on a family of point defects in 2D materials that combine moderate optical oscillator strengths, telecom operation, and low nuclear spin noise. Relative to 3D hosts, 2D hosts provide multiple advantages, including heterostructure engineering[11,12], reduced sensitivity to nuclear spin environment[13], and ease of integration with photonic platforms. Furthermore, the placement of defects in a 2D layer (versus one buried in 3D) could be precisely controlled using a scanning tunneling microscope (STM)[14–16] or focused electron beam lithography[17]. In this regard, defects in monolayer hBN were theoretically investigated as qubit candidates in a 2D host[18–20]. Long, ms-scale, longitudinal spin relaxation times have been demonstrated with a defect ensemble in hBN[21], and optically detected magnetic resonance of single defects in hBN has been reported[22].

[1]Department of Materials Science and Engineering, The University of Texas at Dallas, Richardson, TX 75080, USA. [2]Department of Electronics Engineering, Incheon National University, Incheon 22012, Republic of Korea. [3]Department of Physics, University of California, Santa Cruz, CA 95064, USA. [4]Department of Chemistry and Biochemistry, University of California, Santa Cruz, CA 95064, USA. [5]Department of Physics, University of Washington, Seattle, WA 98195, USA. [6]Department of Electrical and Computer Engineering, University of Washington, Seattle, WA 98195, USA. ✉e-mail: y.lee@inu.ac.kr; kjcho@utdallas.edu

However, spin coherence times (also called quantum memory times) are limited to microseconds in hBN due to the nuclear spin environment (nuclear spins of all B and N isotopes). Ye et al. predicted that a nuclear-spin limited quantum memory time in $MoS_2$ can exceed milliseconds, even considering the natural abundance of nuclear spins before isotopic purification[13]. Moreover, the feasibility of isotopic purification of transition metal dichalcogenides (TMDs) further suppresses decoherence. Despite these promising properties, spin defect qubits in 2D TMDs remain uncharted territory even while defect-based single-photon emitters have been proposed and demonstrated[23–26].

The first-principles calculations based on density functional theory (DFT)[27,28] have extensively contributed to the characterization and identification of defect qubits in a wide range of solid hosts[18–20,29–33]. In this work, we computationally search through defects for a spin defect qubit in 2D monolayer TMDs by means of hybrid DFT[34,35], known to be a quantitatively accurate method for solid-state defect calculations[29,36]. As a result of the comprehensive characterization of fundamental qubit properties—electronic, magnetic, vibrational, optical properties, and thermodynamic stability—we report on a defect family of $M_X$ in monolayer TMDs which turns out to be a promising candidate for quantum network applications.

## Results

### $M_X$ defect family

To computationally discover spin defect qubits realized in monolayer TMDs, we first search through intrinsic (native) and dopant defects in H-MoS, where H is the notation for semiconducting monolayer TMDs[37]. In this initial screening, we consider two criteria: (i) the spin-triplet ground state analogous to the NV center in diamond; (ii) spin-conserving intradefect optical transition without ionization of the defect[29]. High spin states are desirable to decouple the spin from the S = 1/2 paramagnetic background and to allow spin control at zero magnetic field[2]. Spin-conserving optical transitions are required for spin-state readout. First, we conducted DFT calculations based on the Perdew–Burke–Ernzerhof (PBE) functional[38] to quickly explore densities of states for various defect states (Supplementary Fig. 1). As a result of the initial screening in terms of the spin-triplet ground state, we identify three spin-triplet ground-state defect types: negatively charged donor-vacancy complexes ($F_SV_S^{-1}$ and $Re_{Mo}V_S^{-1}$), Mo substitution for two S ($Mo_{S_2}$), and Mo substitution for S ($Mo_S$). The donor-vacancy complexes have the spin-triplet ground state, but their occupied energy levels are not far enough away from the conduction band minimum to avoid ionization of the defects during intradefect optical excitation. Although $Mo_{S_2}$ meets the two criteria, the defect is made of $Mo_I$ and two $V_S$, so $Mo_{S_2}$ is less likely to form than $Mo_S$, which can result in imprecise defect positioning, suffering from a random diffusion process during annealing; furthermore, a sulfur vacancy of $Mo_{S_2}$ locates at the bottom sulfur layer of H-MoS$_2$ does not allow the STM tip manipulation. Out of the initial set of the spin-triplet donor-vacancy, substitution-type defects, we found $Mo_S$ turns out to meet the screening criteria and was selected for a systematic study. We then further characterized the $M_X$ defect family in the semiconducting H-MX$_2$ (M = Mo, W; X = S, Se, Te). WTe$_2$ is excluded because the most stable bulk phase of WTe$_2$ is the metallic T$_d$ phase, not the semiconducting 2H phase[39], and thus unsuitable for hosting an optically active defect. In addition to the criteria above, it is practically desirable that the dopant M is different from the transition metal atoms constituting the host TMD so that we can optically distinguish the synthesized defect qubit from native anti-site defects and reach concentrations low enough for single qubit isolation.

### Defect energy levels

We investigate the detailed electronic structures of selected defects in H-MX$_2$ using hybrid functional DFT calculations[34,35]. The $M_X$ defect family exhibits similar properties, and we focus the main text

discussion on the $W_{Se}$ defect in monolayer MoSe$_2$, which is found to have optical transitions in the telecom band along with $W_S$ in MoS$_2$. The complete data for the family of defects investigated are listed in Table 1. We note that some defects in the family are not suitable spin qubit candidates. For example, $W_{Te}$ in MoTe$_2$ does not have a spin-triplet ground state. For $Mo_{Se}$ in WSe$_2$, the occupied defect levels calculated with spin–orbit coupling (SOC) are lower than the valence band maximum (VBM). Therefore, MoTe$_2$ and MoSe$_2$ can be excluded from the desirable host materials accommodating the $M_X$ defect family. In addition to the $M_X$ defects in monolayer TMDs, Table I includes our simulation results of the NV center in diamond[29] and the $C_BV_N$ defect in monolayer hBN[18,19], which have been reported to meet the aforementioned criteria, although quantum chemistry approaches beyond the hybrid functional demonstrated that the ground state of the $C_BV_N$ in hBN could be spin-singlet by taking into account multi-reference nature of the singlet state[26,40]. These computational results are consistent with the previous reports for the NV center in diamond and $C_BV_N$ in hBN, confirming that our simulation approaches are well-founded for defect qubit predictions. Comparison to these known centers also highlights the distinct features of the $M_X$ defect family.

The structural geometry and spin density of the $N_CV_C^{-1}$, $C_BV_N$, and $W_{Se}$ defects are shown in Fig. 1a–c. All three defects possess the spin-triplet ground states with optical excitation pathways of spin-conserving intradefect transitions. The optical transitions lie within the bandgap $E_g$, prohibiting single-photon ionization of the defect [Fig. 1d–f]; since its estimation based on Kohn–Sham eigenvalues can be erroneous owing to the ambiguous interpretation of the Kohn–Sham eigenvalues, we further confirmed this from the comparison of the zero-phonon line energy and the ionization energy determined by the charge transition level, more precisely (e.g., the zero phonon line energy of $W_{Se}$ in MoSe$_2$ is 0.79 eV, and the ionization energy of that is 1.2 eV). Similar to the NV center in diamond, the $W_{Se}$ defect belongs to the $C_{3v}$ point group, and the electron configuration of the 2D defects is identical to the hole configuration of the NV center. Two majority-spin electrons occupy doubly degenerate $e_x$ and $e_y$ orbitals, and the optical transition takes place between $e_{x,y}$ and $a_1$ orbitals. The quantities between parentheses in Table 1 are given to estimate the SOC effects with heavy elements, where SOC reduces $E_g$ and lifts the degeneracy of the $e_x$ and $e_y$ orbitals. More detailed calculations are required to determine SOC effects on spin coherence times, coherent spin-light interactions, and inter-system crossing[2] and will be addressed in future research work.

### Defect formation energy

Defect formation energy is a crucial quantity to determine whether a proposed defect can be physically realized in a host solid. The defect formation energy of a defect $X^q$ in a charge state $q$ is given by[41–43]

$$E^f[X^q] = E_{tot}[X^q] + E_{corr}^q - E_{tot}[\text{pristine}] - \sum_i n_i\mu_i + q(\epsilon_F + \epsilon_{VBM}^{pristine} - \Delta V_{0/p}) \quad (1)$$

where $E_{tot}[X^q]$ and $E_{tot}[\text{pristine}]$ are the total energies of a supercell with and without the defect $X^q$, respectively. $n_i$ is the number of atoms of a species $i$ added (positive) or removed (negative) from the pristine supercell, $\mu_i$ is the chemical potential of a species $i$. The chemical potential range was determined by considering competing phases (Supplementary Fig. 2) given in phase stability diagrams provided by Materials Project[44]; based on the phase stability diagrams, we further computed the chemical potentials within the HSE06 hybrid functional to plot the formation energy diagrams at extreme conditions, such as the M-rich condition. The chemical potentials of C and N are obtained in the diamond crystal and the N$_2$ molecule, respectively. $E_{corr}^q$ is the electrostatic correction, $\epsilon_F$ is the Fermi level, $\epsilon_{VBM}^{pristine}$ is the VBM energy level in the pristine supercell, and $\Delta V_{0/p}$ is the potential alignment term. The electrostatic correction is employed to take into account

**Table 1 | Summary of calculated defect properties in diamond, hBN, and TMDs**

| Hosts | $E_g$ (eV) | Defects | Point groups | Defect levels (eV) | | | | | | $E_{ZPL}$ (eV) | S | DW | D (GHz) | $\tau_R$ (μs) |
|---|---|---|---|---|---|---|---|---|---|---|---|---|---|---|
| | | | | Spin up | | | Spin down | | | | | | | |
| | | | | $e_x$ | $e_y$ | $a_1$ | $e_x$ | $e_y$ | $a_1$ | | | | | |
| Diamond | 5.47 | $N_C V_C^{-1}$ | $C_{3v}$ | **1.87** | **1.87** | **0.69** | 4.64 | 4.64 | **1.62** | 2.10 | 2.93 | 0.05 | 2.86 | 0.014 |
| hBN | 5.62 | $C_B V_N$ | $C_{2v}$ | **1.53** | **3.00** | **5.30** | 4.76 | 5.05 | 5.77 | 1.71 | 2.21 | 0.11 | 10.77 | 0.032 |
| $MoS_2$ | 2.37 (1.98) | $Mo_S$ | $C_{3v}$ | **0.40 (0.18)** | **0.40 (0.22)** | **1.80 (1.59)** | 2.34 (2.10) | 2.34 (2.10) | 2.35 (2.12) | 1.11 (1.09) | 0.74 | 0.48 | 20.51 | 37.7 |
| | | $W_S$ | $C_{3v}$ | **0.84 (0.48)** | **0.84 (0.71)** | **1.98 (1.89)** | 2.33 (2.04) | 2.33 (2.06) | 2.35 (2.09) | 0.91 (0.94) | 1.47 | 0.23 | 13.44 | 20.5 |
| $WS_2$ | 2.46 (1.85) | $W_S$ | $C_{3v}$ | **0.81 (0.44)** | **0.81 (0.48)** | **2.04 (1.70)** | 2.44 (1.94) | 2.44 (1.97) | 2.46 (1.97) | 1.03 (1.01) | 1.01 | 0.36 | 14.44 | 14.1 |
| | | $Mo_S$ | $C_{3v}$ | **0.38 (0.04)** | **0.38 (0.06)** | **1.92 (1.55)** | 2.44 (1.95) | 2.44 (1.96) | 2.44 (1.97) | 1.22 (1.18) | 0.45 | 0.64 | 21.65 | 54.2 |
| $MoSe_2$ | 2.07 (1.72) | $Mo_{Se}$ | $C_{3v}$ | **0.27 (0.04)** | **0.27 (0.06)** | **1.63 (1.42)** | 2.04 (1.79) | 2.04 (1.79) | 2.08 (1.81) | 1.08 (1.08) | 0.75 | 0.47 | 19.13 | 5.6 |
| | | $W_{Se}$ | $C_{3v}$ | **0.71 (0.29)** | **0.71 (0.52)** | **1.81 (1.57)** | 2.03 (1.75) | 2.03 (1.78) | 2.08 (1.80) | 0.79 (0.74) | 1.95 | 0.14 | 12.43 | 4.2 |
| $WSe_2$ | 2.15 (1.58) | $W_{Se}$ | $C_{3v}$ | **0.69 (0.34)** | **0.69 (0.35)** | **1.84 (1.47)** | 2.10 (1.61) | 2.10 (1.62) | 2.16 (1.66) | 0.88 (0.84) | 1.63 | 0.20 | 12.88 | 3.2 |
| | | $Mo_{Se}$ | $C_{3v}$ | **0.25 (−0.03)** | **0.25 (−0.02)** | **1.74 (1.35)** | 2.12 (1.63) | 2.12 (1.65) | 2.17 (1.66) | 1.19 (1.08) | 0.50 | 0.60 | 19.82 | 4.7 |
| $MoTe_2$ | 1.71 (1.39) | $Mo_{Te}$ | $C_{3v}$ | **0.19 (−0.09)** | **0.39 (0.01)** | **1.26 (0.85)** | 1.66 (1.06) | 1.70 (1.23) | 1.73 (1.30) | 0.49 (0.36) | 4.51 | 0.01 | 0.47 | 7.8 |
| | | $W_{Te}$ | $C_{3v}$ | 1.51 (1.27) | 1.51 (1.24) | **0.86 (0.60)** | 1.51 (1.24) | 1.51 (1.27) | **0.86 (0.60)** | | | | | |

Note: Bold numbers indicate occupied states.

All the values in the table were theoretically estimated in this work, and the numbers between parentheses correspond to results with SOC. Defect levels are relative to the VBM. Majority (minority) spin is referred to as spin up (down). The $e_x$, $e_y$, and $a_1$ columns indicate $a_1$, $b_2$, and $b_2'$ for $C_B V_N$ in hBN, respectively[39]. Note that the notations, $e_x$, $e_y$ and $a_1$, are not valid anymore within SOC.

spurious image charge due to periodic cells and uniform background charge, where the Freysoldt–Neugebauer–Van de Walle (FNV) correction scheme[41–43,45] enables us to handle defects in anisotropic medium, such as 2D materials.

Figure 2 and Supplementary Fig. 4 show that the formation energy of an $M_X$ defect is lower than the sum of formation energies of the two independent defects of $M_I$ and $V_X$ (i.e., $E^f[M_X] < E^f[M_I] + E^f[V_X]$), indicating that the formation of $M_X$ defects is favorable. Compared with the NV center in diamond and $C_B V_N$ in hBN, the formation energy of $M_X$ in TMDs is small, so that the $M_X$ defect family is expected to be readily created. Based on this formation energy, we can create the $M_X$ defects by annealing a system with preexisting $M_I$ and $V_X$ defects. The formation of an antisite defect $Mo_S$ in a $MoS_2$, which is among the $M_X$ defect family, has been confirmed experimentally[46,47]. Along with the experimental observation of $Mo_S$, the similar formation energy diagrams for the $M_X$ defects in the family (Supplementary Fig. 4) support the feasible creation of the $M_X$ defect family. Note that the dopant M needs to be different from transition metal atoms constituting the host TMD to distinguish the intentionally created defect. Since $V_X$ is prevalent in TMDs[48], the $M_X$ defect would be formed near the additional $M_I$ after annealing. Supplementary Figure 6 shows defect formation energies of possible competing defects, where $V_{Se}$ is much easier to be formed than $V_{Mo}$; thus, once we introduce $W_I$ in the presence of abundant $V_{Se}$, the $W_{Se}$ complex can be readily formed. The M atom could be incorporated via ion implantation or STM lithography.

**Zero-phonon line emission**
Photon emission of defects plays a key role in qubit operation. Spin-conserving cycling transitions are utilized to read out the spin-qubit state. Zero-phonon-line (ZPL) transitions are utilized to realize spin-photon entanglement, which is required for generating spin-entangled quantum networks via photon measurement[49,50]. The ZPL emission is also utilized as a spectroscopic fingerprint to identify the defect qubit[2,31,32]. A photoluminescence line shape is composed of the ZPL and phonon sidebands. The contribution of the ZPL emission to the total emission is estimated by the Debye-Waller (DW) factor[31,32]. Only the ZPL emission is useful for photon-spin entanglement schemes, and thus a high DW factor is desirable. The configuration coordinate diagram (adiabatic potential energy against configuration coordinate) is often utilized to investigate the ZPL emission (Fig. 3)[51,52]. Here, the configuration coordinate displacement $\Delta Q$ is calculated as[51,52]

$$\Delta Q = \sqrt{\sum_{\alpha,i} m_\alpha (R_{e;\alpha i} - R_{g;\alpha i})^2}, \tag{2}$$

where $i = \{x, y, z\}$, $m_\alpha$ is the mass of atom $\alpha$, $R_{g(e);\alpha i}$ is the equilibrium position in the ground (excited) state. The electronic excited state is calculated by using the constrained DFT[53]. The number of phonons emitted during the optical transition can be quantified by the Huang–Rhys factor $S$. In the one-dimensional (1D) effective phonon approximation[51], $S = \frac{\Delta E}{\hbar\omega}$, where $\Delta E$ and $\hbar\omega$ are described in Fig. 3a, and the effective phonon frequency $\omega$ is obtained using the harmonic oscillator approximation $E = \frac{1}{2}\omega^2 Q^2$. Here, $\Delta E$ is the difference between the ZPL energy $E_{ZPL}$ and the vertical emission energy. The DW factor is given by $DW = e^{-S}$ [31,32,52]. The ZPL energies, the Huang–Rhys factors, and the DW factors for the $M_X$ defect family are shown in Table 1. The $M_X$ defect family exhibits larger DW factors than the NV center in diamond and the $C_B V_N$ in hBN except for in the $MoTe_2$ host (which is not promising from the energy level point of view, as discussed earlier). The large DW factors stem from the small curvature of the $M_X$ defect family configuration coordinate diagram compared with the NV center in diamond and the $C_B V_N$ in hBN (Fig. 3). In Table 1, the SOC-corrected ZPL energies between parentheses are approximated by estimating shifts in the defect energy levels shown in the same table. The ZPL energies of the $M_X$ defect family typically lie around 1 eV, close or in the

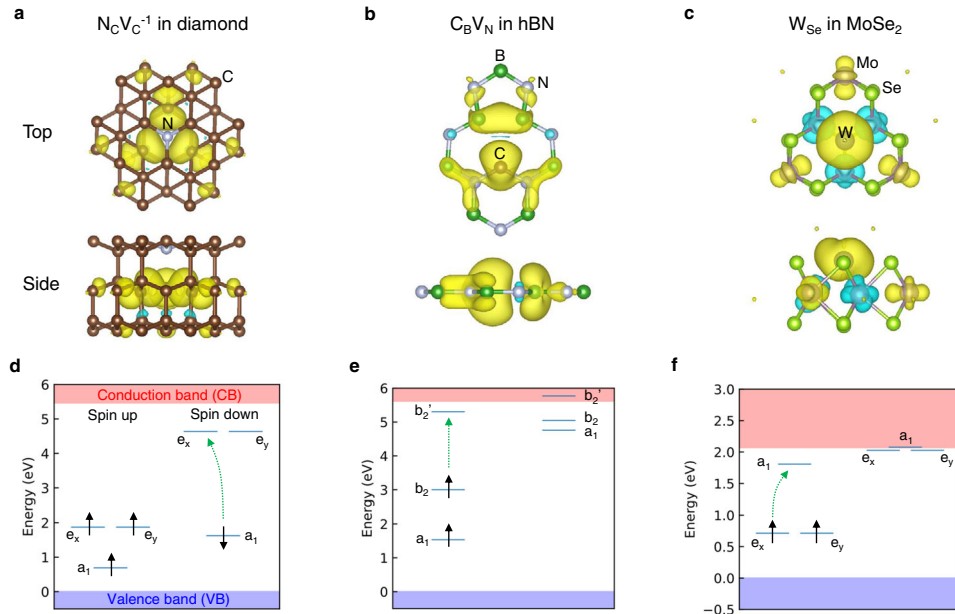

**Fig. 1 | Defect geometries and calculated electronic structures of (a,d) $N_C V_C^{-1}$ in diamond, (b, e) $C_B V_N$ in hBN, and (c,f) $W_{Se}$ in MoSe₂. a–c** Top (top) and side (bottom) views of defect geometries and spin densities of the defect qubits in the ground state (isosurface level = 0.003 Å⁻³). **d–f** Energy levels of the defect qubits.

The green arrows indicate spin-conserving intradefect optical transition. Detailed physical quantities of possible combinations of $M_X$ defects and $MX_2$ hosts are summarized in Table 1.

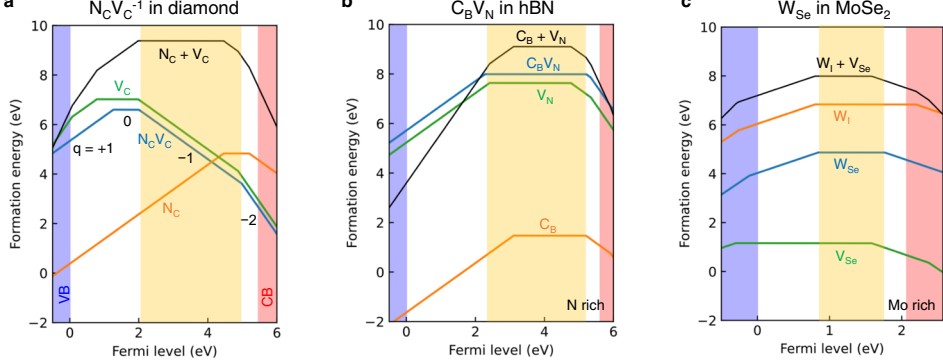

**Fig. 2 | Defect formation energy diagrams.** Defect formation energies of the defect qubits for **a** $N_C V_C^{-1}$ in diamond, **b** $C_B V_N$ in hBN in N-rich condition, and **c** $W_{Se}$ in MoSe₂ in Mo-rich condition. The M-rich condition for $MX_2$ (N-rich condition for hBN) provides lower defect formation energies than the X-rich condition (B-rich condition) (Supplementary Fig. 3). **c** $W_I + V_{Se}$ indicates the sum of formation energies of the two independent defects, $W_I$ and $V_{Se}$. The orange-shaded area shows the range of stability of $N_C V_C^{-1}$, $C_B V_N^0$, and $W_{Se}^0$. Defect formation energy diagrams for other $M_X$ in TMDs are displayed in Supplementary Fig. 4.

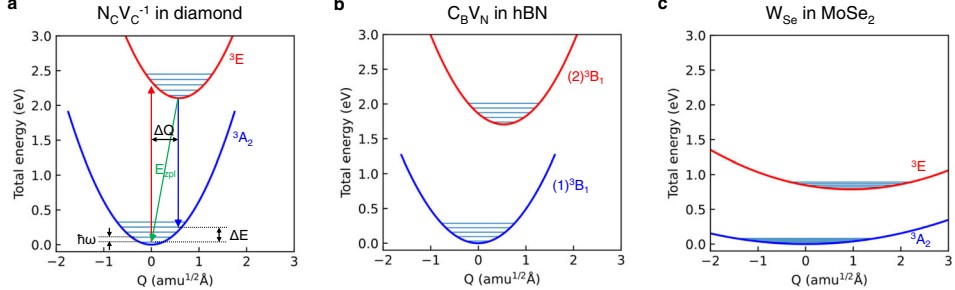

**Fig. 3 | Configuration coordinate diagrams responsible for ZPL emission.** Configuration coordinate diagrams of the defect qubits for **a** $N_C V_C^{-1}$ in diamond, **b** $C_B V_N$ in hBN, and **c** $W_{Se}$ in MoSe₂. The solid horizontal lines correspond to the

phonon energy levels in the harmonic approximation. The ground state and the excited state are labeled as ³A₂ and ³E for $N_C V_C^{-1}$ in diamond[31] and $M_X$ in TMDs; the states are labeled as (1)³B₁ and (2)³B₁ for $C_B V_N$ in hBN[40].

telecom band, with the SOC-corrected ZPL energies at 0.74 eV and 0.94 eV of the $W_{Se}$ in $MoSe_2$ and $W_S$ in $MoS_2$, respectively. As we show further below, the 2D host environment enables fine-tuning of the ZPL energy by applying strain to further minimize optical fiber transmission loss. While photoluminescence measurements in 2D TMDs have identified localized excitons and chalcogenide vacancies[54–56], a ZPL that can be attributed to the $M_X$ defect family has not been experimentally identified yet.

## Zero-field splitting and hyperfine tensors

Magnetic properties of a spin qubit are of paramount importance for realizing quantum information applications[2,31,32]. For instance, the zero-field splitting (ZFS) between the $m_s = 0$ and $m_s = \pm 1$ spin sublevels corresponds to the microwave energy to manipulate the qubit state at zero applied magnetic field and enables spin-selective resonant optical excitation. Furthermore, utilizing ZFS TMD-based defects may also be attractive for quantum sensing applications. In these applications, changes in the ZFS can be used to sense electric fields, strain, and temperature[57,58], while splitting of the $m_s = \pm 1$ state due to the Zeeman effect is used for magnetic sensing applications[59,60]. Also important is the magnetic coupling of the spin-qubit to the crystal spin bath. A ZFS allows one to decouple the qubit spin from a paramagnetic electron spin-1/2 bath. In addition to a qubit spin state coupling to paramagnetic electron spins, there will also be hyperfine couplings to the crystal host nuclear bath[13]. In spin Hamiltonian, the ZFS and the hyperfine interaction are described as $\sum_n \hat{\mathbf{S}}^T \cdot \mathbf{A}^{(n)} \cdot \hat{\mathbf{I}}^{(n)}$ and $\hat{\mathbf{S}}^T \cdot \mathbf{D} \cdot \hat{\mathbf{S}}$, respectively, where $\hat{\mathbf{S}}$ is the electron spin, $\hat{\mathbf{I}}^{(n)}$ is the nuclear spin of nucleus $n$, $\mathbf{D}$ is the ZFS tensor, and $\mathbf{A}^{(n)}$ is the hyperfine tensor.

The ZFS tensor determines the dipolar spin–spin interaction between electrons and is given by[32,61]

$$D_{ab} = \frac{1}{2} \frac{\mu_0}{4\pi} \frac{g_e^2 \mu_B^2}{S(2S-1)} \sum_{i>j}^{\text{occupied}} \chi_{ij} \left\langle \Psi_{ij}(\mathbf{r}_1, \mathbf{r}_2) \Big| \frac{r^2 \delta_{ab} - 3r_a r_b}{r^5} \Big| \Psi_{ij}(\mathbf{r}_1, \mathbf{r}_2) \right\rangle, \tag{3}$$

where $\mu_0$ is the magnetic permeability of vacuum, $g_e$ is the electron g-factor, $\mu_B$ is the Bohr magneton, $\chi_{ij}$ is +1 for parallel spins and −1 for antiparallel spins, $r_{a,b} = (\mathbf{r}_1 - \mathbf{r}_2)_{a,b}$, and $|\Psi_{ij}(\mathbf{r}_1, \mathbf{r}_2)\rangle$ is the slater determinant of $i$th and $j$th Kohn-Sham orbitals. After diagonalizing the ZFS tensor, one can obtain the ZFS value $D = \frac{3}{2}D_{zz}$, presented in Table 1 (see Supplementary Table 1 for the tensor elements, $D_{xx}$, $D_{yy}$, and $D_{zz}$). For the NV center in diamond, $D$ calculated in this work is 2.86 GHz, which is close to the reported one[2,32]. $D$ of the $M_X$ defect family are 10–20 GHz, about an order of magnitude larger than that of the NV center, which is within the experimentally accessible range of microwave control[62,63] and could enable higher-temperature resonant spin readout as well as the compatibility of higher Purcell factors[64] with resonant optical spin selectivity. The large $D$ of the $M_X$ defect family is attributed to stronger dipolar spin-spin interaction due to the more localized electron wavefunctions than the NV center. Note that because of the additional contribution of SOC[65], the ZFS could be even greater than the value presented in Table 1, especially with a heavy element, such as W.

The hyperfine tensor of nucleus $n$ at $r = 0$ is calculated by using[32,66]

$$A_{ab}^{(n)} = \frac{\mu_0}{4\pi} \frac{g_e \mu_B g_n \mu_n}{S} \int d^3 r\, n_s(\mathbf{r}) \left[ \left( \frac{8\pi}{3} \delta(r) \right) + \left( \frac{3r_a r_b}{r^5} - \frac{\delta_{ab}}{r^3} \right) \right], \tag{4}$$

where $n_s(\mathbf{r})$ is the electron spin density [Fig. 1(a–c)], $g_n$ is the nuclear g-factor[67], and $\mu_n$ is the nuclear magneton. In Eq. (4), the first parenthesis is the non-dipolar Fermi contact term, and the second parenthesis is the dipole–dipole interaction term. Table 2 displays the calculated and diagonalized hyperfine tensors of the NV center in diamond, $C_B V_N$ in hBN, and $W_{Se}$ in $MoSe_2$ (see Supplementary Table 2

for a full list of hyperfine tensors, including other $M_X$ in the family) at the defect and nearest neighbor sites. The $^{183}W$ and $^{77}Se$ nuclear spins of the $W_{Se}$ defect exhibits large hyperfine tensor elements, similar to the on-site interaction $C_B V_N$ defect in hBN and the nearest neighbor $^{13}C$ in the NV. Considering the number of equivalent sites, the total hyperfine coupling between the electron spin and nearby nuclear spins is not necessarily stronger than the NV center. Furthermore, the advantageous dimensionality[13] and the isotopic purification for 2D TMDs are expected to provide an exceptionally coherent time, whereas 2D hBN is incapable of excluding spinful nuclear isotopes. One intriguing possibility with 2D TMDs is to completely engineer a nuclear spin quantum memory register[68] by STM lithography[15]. In this case, one would begin with an isotope-purified spin-0 host and incorporate a handful of nonzero spin nuclei in proximity to the defect.

## Radiative decay

In addition to the DW factor, the radiative recombination rate is an important optical property. For quantum information protocols, recombination rates should be fast enough to realize efficient spin initialization and readout[2,31,32]. Practically, radiative rates should also exceed the rates of any nonradiative recombination processes. The radiative recombination rate, which is the inverse of the radiative recombination lifetime $\tau_R$, is calculated using[32,69]

$$\frac{1}{\tau_R} = \frac{n E_{ij}^3 |\mu_{ij}|^2}{3\epsilon_0 \pi c^3 \hbar^4}, \tag{5}$$

where $n$ is the refractive index, $\epsilon_0$ is the vacuum permittivity, $E_{ij}$ is the excitation energy that is substituted with $E_{ZPL}$, and $\mu_{ij} = \langle \psi_j | e\mathbf{r} | \psi_i \rangle$ is the transition dipole moment between the initial state $|\psi_i\rangle$ and the final state $|\psi_j\rangle$. Under the Frank-Condon approximation, we consider only the electronic component of the initial and final wavefunctions, which are occupied and empty Kohn-Sham orbitals of the spin-triplet ground state. Table 1 shows the calculated $\tau_R$ for the systems that we have examined so far. The $W_{Se}$ in $MoSe_2$ exhibits a 4.2 μs decay time, which is four times shorter than the 20.5 μs decay time of the $W_S$ in $MoS_2$. Overall, $\tau_R$ of the $M_X$ defect family is 100–1000 times larger compared with the NV center in diamond and $C_B V_N$ in hBN. In the $M_X$ defect family, the optical transition between $e_{x,y}$, and $a_1$ is smaller due to the orbital selection rule (Laporte rule)[70] associated with distinct $d$ orbital characters of their defect states, $e_x$, $e_y$, and $a_1$ (Supplementary Figure 6). While (slightly) shorter $\tau_R$ may be desirable, we note $\tau_R$ is already 5 orders of magnitude shorter than the current most promising defect telecom qubit, $Er$:$^{3+}Y_2SiO_5$ where the intra-f-shell transitions are utilized, unlike the transition metal defects with d-orbital physics[6]. Moreover, for efficient photon collection, cavity integration is required, which can reduce $\tau_R$ by 4 orders of magnitude via the Purcell effect[6]. Due to the large ZFS, the system should still retain frequency-selective spin excitation for spin-photon entanglement and spin readout even with the 4 orders of magnitude frequency broadening. TMDs can also provide multiple advantages in sensing. Due to the proximity to the surface, the exposed defect qubit on the surface of monolayer TMDs can compensate for the low radiative decay rate by suppressing internal reflection. Together with the radiative process, nonradiative recombination is a vital process determining quantum yield. The absence of crossing between the potential energy curves of $^3E$ and $^3A_2$ shown in Fig. 3c indicates that the nonradiative transition between the triplet states is less likely to occur; however, further investigation is necessary to make sure the rare nonradiative transition because the transition could depend on many critical factors.

## Intersystem crossing (ISC)

The transition between a triplet state and a singlet state can play an important role in a nonradiative process and can enable the low-fidelity room-temperature optical initialization and readout of the

**Table 2 | Calculated hyperfine tensors for $N_CV_C^{-1}$ in diamond, $C_BV_N$ in hBN, and $W_{Se}$ in $MoSe_2$**

| Hosts | Defects | Nuclear spins | Number of equivalent sites | Hyperfine tensors (MHz) (convention: $\|A_{zz}\| > \|A_{xx}\| > \|A_{yy}\|$) | | |
|---|---|---|---|---|---|---|
| | | | | $A_{xx}$ | $A_{yy}$ | $A_{zz}$ |
| Diamond | $N_CV_C^{-1}$ | $^{14}$N ($I = 1$, 99.632%) | 1 | −2.9 | −2.6 | −2.9 |
| | | $^{15}$N ($I = 1/2$, 0.368%) | 1 | 4.1 | 3.6 | 4.1 |
| | | $^{13}$C ($I = 1/2$, 1.07%) | 3 | 145.0 | 144.8 | 227.2 |
| | | $^{13}$C ($I = 1/2$, 1.07%) | 6 | 14.2 | 14.1 | 19.9 |
| hBN | $C_BV_N$ | $^{13}$C ($I = 1/2$, 1.07%) | 1 | 474.7 | 400.9 | 478.8 |
| | | $^{10}$B ($I = 3$, 19.9%) | 1 | 24.9 | 22.2 | 26.4 |
| | | $^{11}$B ($I = 3/2$, 80.1%) | 2 | 74.4 | 66.3 | 78.9 |
| | | $^{14}$N ($I = 1$, 99.632%) | 2 | 7.3 | 7.2 | 9.9 |
| | | $^{15}$N ($I = 1/2$, 0.368%) | 2 | −10.3 | −10.1 | −13.9 |
| $MoSe_2$ | $W_{Se}$ | $^{183}$W ($I = 1/2$, 14.31%) | 1 | 332.9 | 253.0 | 333.0 |
| | | $^{95}$Mo ($I = 5/2$, 15.92%) | 3 | 14.6 | 8.4 | 16.5 |
| | | $^{97}$Mo ($I = 5/2$, 9.55%) | 3 | 14.9 | 8.6 | 16.8 |
| | | $^{77}$Se ($I = 1/2$, 7.63%) | 6 | 68.2 | 65.4 | 78.2 |

The nuclear spin quantum number $I$ and natural abundance are displayed in the nuclear spins column.

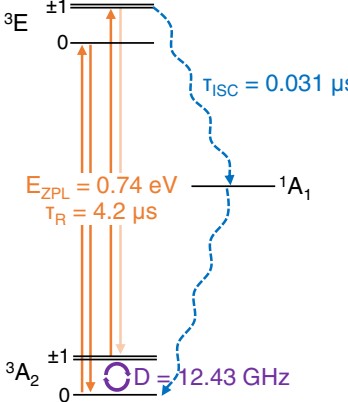

**Fig. 4 | Sublevel structure of $W_{Se}$ in $MoSe_2$.** The radiative processes are shown in the orange vertical line. The blue dashed lines show the symmetry-allowed ISC transitions from the triplet excited state $^3$E to the singlet state $^1A_1$ and the transition from $^1A_1$ to $^3A_2$, which are responsible for spin-selective decay, enabling the initialization and readout operations. The purple circular arrows within ZFS indicate the manipulation of qubit states by microwave.

qubit-based sensors. The $M_X$ defect family symmetrically resembles the antisite defect in monolayer TMDs and is expected to exhibit symmetry-allowed ISCs as in the antisite defect[71]. ISC is mediated by a combination of SOC and electron-phonon interaction. The crossing rate was calculated by the application of Fermi's golden rule according to the formula[72,73]:

$$\Gamma_{ISC} = 4\pi\lambda_\perp^2 \tilde{X}_{if}, \quad (6)$$

$$\tilde{X}_{if} = \sum_m w_m \sum_n \left| \left\langle \phi_{im} | \phi_{fn} \right\rangle \right|^2 \delta\left( \Delta E_{if} + m\hbar\omega_i - n\hbar\omega_f \right), \quad (7)$$

where $\lambda_\perp$ is the transverse SOC constant between spin-singlet and spin-triplet states, $\tilde{X}_{if}$ is the phonon wavefunction overlap between initial state $i$ with phonon quantum number $m$ and final state $f$ with phonon quantum number $n$, $\phi_{im}$ and $\phi_{fn}$ are the phonon wavefunctions, $\omega_i$ and $\omega_f$ are the phonon frequencies, $w_m$ is the occupation number of phonon according to Bose-Einstein distribution, and $\Delta E_{if}$ is the energy difference between the initial state and final state (See Methods for

further details of phonon wavefunction overlap and SOC strength calculations). The ISC from the triplet excited states $^3$E to the singlet shelving state $^1A_1$ can be symmetrically allowed when $m_s = \pm 1$[71]. The simulated transition rate of ISC from the triplet excited state to the singlet shelving state is 0.031 μs, which is shorter than the radiative lifetime 4.2 μs of the triplet excited state, which tells us that the proposed quantum defect can exhibit the initialization and readout operation via the spin-selective decay pathways (Fig. 4). We note, however, that for the high-fidelity initialization and readout required for computation and network, resonant, spin selective excitation is required along with avoided or minimized ISCs[74]. Since SOC underlies the ISC transition[72], we will be able to engineer ISC by utilizing various transition metal dopants with different SOCs.

**Strain engineering**

Strain can be effective in altering the dominant $d$ orbital character by reducing the defect-crystal symmetry, which significantly modulates defect qubit properties, including the optical transition properties. Therefore, we can modify the radiative recombination rate under applied strain. As shown in Fig. 5, uniaxial strain along $x$ or $y$ breaks the $C_{3v}$ symmetry and lifts the $e_x$ and $e_y$ degeneracy. Technically, the notation $e_{x,y}$ is not valid when uniaxial strain is applied, and the notation is associated with their original orbital without strain. Although the biaxial strain does not break the $C_{3v}$ symmetry, the biaxial strain affects orbital mixing, resulting in the modulation of $\tau_R$ and $E_{ZPL}$. If we consider the lowest excitation for qubit operation ($e_x \to a_1$ for uniaxial strain along $x$, $e_y \to a_1$ for uniaxial strain along $y$), uniaxial strain is always beneficial to achieve a shorter lifetime. The tensile biaxial strain would also be helpful. In addition to $\tau_R$, the strain technique can be used to engineer the ZPL energy. As shown in Fig. 5d–f, strain shifts energy levels and changes the energy gaps between $a_1$ state and $e_{x,y}$ states by a few hundred meV, which would provide a useful way to tune defects to a single operational frequency in a targeted communication band. 2D host materials are beneficial for the strain engineering of defect qubit properties because a single atomic sheet can accommodate a more significant mechanical strain (up to a few %) than bulk materials (typically less than 0.1%), and the strain will depend on the 2D material-substrate interfaces[75]. Interestingly, $\delta E$ pertaining to the second-lowest energy excitation, changes abruptly with small uniaxial strains as a consequence of the lifted degeneracy due to the symmetry breaking. The drastic response to external strain could be promising for highly susceptible quantum strain sensors.

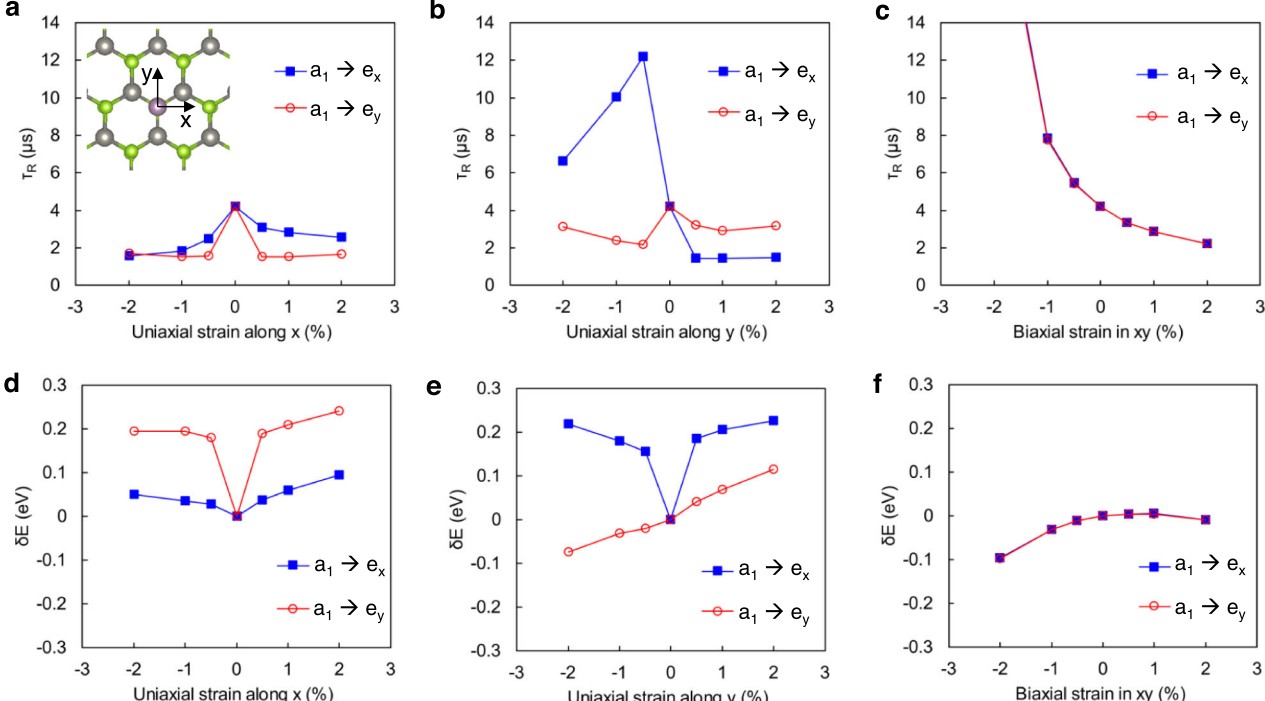

**Fig. 5 | Strain effects on radiative recombination lifetime and ZPL energy.** **a**, **b** Uniaxial and **c** biaxial strain effects on radiative recombination lifetime $\tau_R$. The inset in **a** shows the directions $x$ and $y$. **d**, **e** Uniaxial and **f** biaxial strain effects on $\delta E$, modulation of the gap between corresponding eigenvalues ($a_1$ to $e_{x,y}$) in the ground state. The $\delta E$ approximates the change of the ZPL energy, assuming the vertical shift of adiabatic potential energy curves occurs in the configuration coordinate diagram.

## Discussion

We proposed the $M_X$ defect family in monolayer TMDs as a promising solid-state defect qubit through systematic computational investigation of essential criteria: defect energy levels, defect formation energy, ZPL emission, ZFS, hyperfine tensor, radiative recombination rate, and ISC transition rate. Compared with the NV center in diamond and the $C_BV_N$ defect in hBN, the proposed defects exhibited desirable qubit properties, operating at telecom wavelengths. Finally, we demonstrated strain effects on radiative recombination lifetime and defect energy levels, which provides a technique that we can exploit for further engineering qubit properties and applications to sensitive quantum strain sensors.

Among the various combinations of M and X, the $W_{Se}$ defect in $MoSe_2$ and the $W_S$ defect in $MoS_2$ are particularly promising candidates for quantum network applications with a ZPL transition in the telecom band. However, many of the family's defects are promising candidates for the first demonstration of experimentally detected spin defect qubits in a 2D TMD host. Computationally, there is also further work to be performed. In particular, the role of spin-orbit coupling on spin $T_1$ lifetimes and coherent spin-light interactions should be investigated. The Debye temperatures of TMDs[76] are an order of magnitude smaller than those of the NV center in diamond and the $C_BV_N$ defect in hBN; thus, it is reasonable to expect that spin relaxation time $T_1$ of the $M_X$ defect family could be shorter than those of the counterparts due to a strong spin-phonon interaction[3,77]. If $T_2$, such that $T_2 \leq 2T_1$, is limited by $T_1$, one can explore different combinations of defects and hosts in the family to mitigate the spin-phonon interaction by reducing SOC. Other transition metal atoms in adjacent columns of the periodic table can also be explored to substitute for X along with a nonzero change state, implying expansive room for further exploration and qubit property engineering of 2D quantum defect systems. Having theoretically discovered and characterized the promising spin-defect qubits in monolayer TMDs, we opened a new door to the 2D world of research on spin-defect qubits.

## Methods
### First-principles calculations

We used Vienna Ab initio Simulation Package (VASP)[78,79] to perform the first-principles calculations based on density functional theory (DFT)[27,28]. The Heyd–Scuseria–Ernzerhof (HSE06) hybrid functional[34,35], partially incorporating the Hartree-Fock exchange interaction, is used to overcome the bandgap problem with local exchange-correlation functionals. The pseudopotential is given by the projector-augmented wave method[80,81]. The energy cutoff for the plane-wave basis set is 250 eV for monolayer TMDs (350 eV for diamond and monolayer hBN). We prepared a supercell of $6 \times 6 \times 1$ primitive cells for pristine monolayer TMDs and hBN ($3 \times 3 \times 3$ cubic unit cells for diamond), including a 15-Å-thick vacuum region. The single $\Gamma$-centered $k$-point is adopted for the Brillouin zone sampling. A pristine cell geometry is optimized until the maximum atomic force is smaller than 0.02 eV/Å; then, a defective cell geometry is relaxed within a fixed cell shape and volume based on the optimized pristine cell. The SOC is not considered unless otherwise stated. We utilized subroutines implemented in VASP to compute the magnetic properties—the ZFS tensors and the hyperfine tensors. We used the Corrections For Formation Energy and Eigenvalues (CoFFEE) code[43] to calculate defect formation energies with the FNV charge correction scheme[42].

### Phonon wavefunction overlap and SOC strength

ISC is attributed to a combination of SOC and electron-phonon interaction. To obtain the phonon wavefunction overlap between the initial and final state, a one-dimensional harmonic oscillation approximation was used, which introduces the general configuration coordinate diagram. The potential surfaces of spin-triplet excited state $^3E$ and spin-singlet state $^1A_1$ were obtained by linearly interpolating between initial $^3E$ and final $^1A_1$ structures involved in the ISC. Energies of the interpolated structure were calculated using constrained-occupation DFT[73]. Since Kohn-Sham DFT theory cannot describe states composed of multiple Slater determinates, approximate electron occupations—

$|a_1 e_x\rangle$ for $^3$E and $|e_x \bar{e}_y\rangle$ for $^1$A$_1$—were adopted, where $\bar{e}_y$ indicate the different spin channel of $e_y$ orbital, and we made an approximation to access the energy of the $^1$A$_1$ at the equilibrium geometry following Mackoit-Sinkeviciene et al.[82]. All constrained DFT computations were performed using VASP, facilitated by modified Nonrand[83] preprocessing and postprocessing for interpolated structure energy calculation. The calculated configuration coordinate diagram for $^3$E and $^1$A$_1$ is shown in Supplementary Fig. 7.

SOC strength was computed with the ORCA code[84] using time-dependent density functional theory[85]. Different from VASP, ORCA does not have the feature of periodic boundary conditions. We thus constructed cluster models for both N$_C$V$_C^{-1}$ and W$_{Se}$ defects by cutting relaxed structures from bulk and saturating dangling bonds to reproduce the electronic structures of bulk structures. The dangling bonds in the diamond cluster are easily saturated by H, while TMD is well-known for complicated edge states and charge transfer between edges and defects for over 10 Å[86]. After testing with different sizes, boundaries, and termination groups, a cluster with hybrid zigzag and armchair boundary and termination groups of H, OH, and NH was found using B3LYP functional to have both the same spin density as a periodic result [Supplementary Fig. 8a, b] and HOMO-LUMO gap of 1.22 eV to get reasonably excited states [Supplementary Figure 8(c)]. We obtained SOC values of 4.71 GHz for $\lambda_\parallel$ and 44.6 GHz for $\lambda_\perp$ for N$_C$V$_C^{-1}$ defect using PBE functionals with def2-TZVP basis, which agrees well with previously computed values and experimentally measured values[72,73,87]. With the calculated $\lambda_\perp$, we obtained the $^3$E → $^1$A$_1$ ISC rate for the NV center in diamond at 30.6 MHz which is in fair agreement with the literature-reported value of 60.7 MHz[88]. We then computed the SOC strength for the axial $\lambda_\parallel$ and non-axial $\lambda_\perp$ components of the W$_{Se}$ defect in MoSe$_2$ using B3LYP functionals to be 69 and 109 GHz, respectively.

## Data availability

The data that support the findings of this study are available within the paper and Supplementary Information. Additional relevant data are available from the corresponding authors upon reasonable request.

## Code availability

The codes used for data acquisition and processing are available from the corresponding authors upon reasonable request.

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

## Acknowledgements

This work was supported by ASCENT, one of six centers in JUMP, a Semiconductor Research Corporation (SRC) program sponsored by DARPA (Grant no. 2018-JU-2776). This work was also supported by UTD Quantum Center, UTD SPIRe Award, National R&D Program (2022M3H4A1A04096496) and Basic Science Research Program (Grant no. 2022R1F1A1073068) through the National Research Foundation of Korea (NRF) funded by Ministry of Science and ICT. K.-M.C.Fu acknowledges support from the Army Research Office MURI (Ab-Initio Solid-State Quantum Materials) Grant no. W911NF-18-1-0431. Y.P. acknowledges support from the National Science Foundation under Grant no. DMR-2143233.

## Author contributions

K.C. conceived the research idea and supervised the overall project development. Y.L. developed the idea, performed the first-principles calculations, and analyzed data. Y.H., X.L., K.L., and Y.P. performed the ISC calculation. Y.L., D.K., K.-M.C.F., and K.C. contributed to the discussion and preparation of the first draft of the paper.

## Competing interests

The authors declare no competing interests.
