## [Peer Review File · Nature Communications]

REVIEWER COMMENTS

Reviewer #1 (Remarks to the Author):

Authors performed density functional theory (DFT) calculations to search for a new defect for qubit generation, especially at telecom wavelength. Some qualitative (not quantitative) criteria were selected, and because of that generalized gradient approximation calculations were able to be used to screen some of the defects at the first stage. After that hybrid DFT calculations were performed to further analyze the electronic structure quantitatively. For a few selected defects, more physical properties were investigated deeply. I have no doubt that the calculations were performed with care, however, there is one thing that must be clarified before the final decision.

The formation energy of defects is too high. For instance, the NV center has defect formation energy of more than 9 eV in the shaded region in Figure 2. The proposed W_{Se} defect also has high formation energy around 4 eV. This is really reasonably low value to form the defects in 2D materials? In other semiconductors, as far as I know, such defects are not considered to be formed.

Reviewer #2 (Remarks to the Author):

Y. Lee and the coauthors reported first-principles density functional calculations on spin defects in TMDC materials to theoretically search for suitable spin qubit candidates, which could be optically addressable. The authors found that substitutional metal impurity for X in MX₂ TMDC hosts would produce spin-triplet ground state along with spin-conserving optical transition energy in the telecom range. Hybrid density functional calculations were employed to predict the basic single-electron band structure and the defect formation energy of the suggested defects. The authors also computed spin Hamiltonian parameters such as zero-field splitting and hyperfine parameters, and radiative recombination rates using DFT.

The motivation of this work is interesting and timely. A search of spin qubits in a 2D materials host is an important research topic and it gains a large amount of attention as correctly pointed out by the authors. However, I found that this theory-only work does not support enough their main claim that the MX defect is a spin qubit candidate in the TMDC materials. Furthermore, I found several other critical issues in the manuscript and the work does not meet the expected standards for high-profile journals like nature communications. Therefore, I do not recommend the publication of this work at Nature Communications. In the following, I summarize some critical issues to be addressed. After

making the following changes and downgrading some of their strong claims, the authors should submit it to a physics journal, which better suits the overall scope of this work.

1. There are no results and discussions on singlet shelving states, which play a crucial role in optical initialization and readout of NV-like spin qubit systems. The authors should compute the full many-electron energy levels of the suggested defects and show compelling theoretical evidence of intersystem crossing which could lead to optical initialization in order to fully support their main claim.

2. Calculation and estimation of the non-radiative lifetime would be highly desirable to examine the possibility of the intersystem crossing. Without considering non-radiative intersystem crossing rates, it is impossible to imagine that this system can function as NV-like optically addressable spin qubits.

3. The suggested defects have very large ZFS parameters (~ 10 GHz), so it is not practically feasible to realize spin qubits by using these defects because of experimental constraints.

4. The authors should present a detailed comparison between the zero-phonon line and the ionization energy of the defects. The authors computed the charge transition levels, so it is possible to compute the ionization energy by considering the bandgap of the defects. This is required to support their claim that the defects would not be ionized during optical excitation.

5. The ground-state of CBVN in h-BN is not spin-triplet but spin-singlet. Please consider the latest literature on the h-BN quantum defects and take it into account.

6. The authors claim that the MX defects could be readily created because their defect formation energy is smaller than that of the diamond NV center or that of CBVN in h-BN, and it is also lower than the sum of formation energies of the MI and VX. But, these are not enough to support their claim. First of all, defect formation mechanism and kinetics in h-BN ($E_g \sim 6$ eV) or in diamond ($E_g \sim 5$ eV) are completely different from those in TMDC materials ($E_g \sim 1 \sim 2$ eV). So the mere comparison between the defect formation energies is not enough to claim the experimental feasibility of creating MX defects in TMDC materials. Second, the authors should compare the defect formation energy of the MX defects to that of other well-known defects or possible competing defects. For example, would MI defects are stable enough, or was it experimentally observed? How is the defect formation energy of MX compared to that of M on the metal site?

7. It seems that the authors used the HSE06 functional for all the MX₂ host materials considered in this study without adjusting the mixing parameter and screening parameter. Is this choice good enough to accurately predict the band-gap of all materials, which is an important quantity particularly for this study?

The followings are other minor issues.

1. In table 1, the authors reported the band gaps. Are these theoretical band gaps? What are the numbers in parentheses? Please describe them in detail.

2. Please describe how the chemical potential range for the atomic elements was computed when computing the defect formation energies. What competing phases were considered for W, Mo, etc?

3. On page 6, the authors stated that "The optical transitions lie within the bandgap E_g , prohibiting single-photon ionization of the defect [Figure 1(d-f)].", which is not true. The optical transition energy should be compared to the ionization energy of the defect.

4. On page 16, the authors stated "The WSe in MoSe₂ exhibits a 4.2 μ s decay time. Overall, τ_R of the MX defect family is 100-1000 times larger compared with the NV center in diamond and CBVN in hBN. While (slightly) shorter τ_R may be desirable, we note τ_R is already 5 orders of magnitude shorter than the current most promising defect telecom qubit, Er³⁺:Y₂SiO₅". But I think this is not a fair comparison. The authors claimed that the MX defects would be spin qubit candidates similar to the NV center in diamond. Then, it looks strange to compare its optical property to Er³⁺:Y₂SiO₅, which operates in a completely different way.

5. On page 16, the authors stated "With moderate τ_R and a large ZFS, it is possible to achieve this enhancement while still retaining frequency-selective spin excitation for spin-photon entanglement and spin read-out." But, there is no support for this claim. So, this sentence should be removed.

6. On page 16, the authors stated "Together with the radiative process, nonradiative recombination is a vital process determining quantum yield. The absence of crossing between the potential energy curves of 3E and 3A₂ shown in Figure 3(c) indicates that the nonradiative transition between the triplet states is unlikely to occur." This is also not right. The nonradiative transition between the

triplet states depends on many critical factors. Without considering them, the authors should not claim like this.

Reviewer #3 (Remarks to the Author):

In this manuscript, Lee et al. present a first-principles study of a novel family of spin-defects in TMDs. The authors reported various thermodynamic, electronic, optical, and magnetic properties of these defects obtained from DFT calculations. Overall, I think this work is sound and interesting from a first-principles perspective, but I am not convinced of the experimental relevance of the defects proposed by this work. Therefore, I think this work is more suitable for a more specialized journal.

My major comments are:

1. The defect formation energy reported in Fig. 2c seems to be relatively high for the new defect. I am not sure if it is feasible to create the defects experimentally. Without experimental evidence or convincing theoretical arguments, the existence of the defects proposed by the paper is only hypothetical.
2. From Table 1, it seems including the SOC effect significantly shifts the DFT results for energy levels, but the calculation of ZPL does not include the SOC effect. It is unclear whether the new defects would still be predicted to operate on telecom wavelengths if the SOC effect is taken into account.
3. The authors claim that "The ZFS tensor determines the dipolar spin-spin interaction between electrons". This is not correct. In fact, the spin-spin interaction is only one contribution to the ZFS tensor. For main group systems like diamond or hBN, the spin-spin interaction is usually the dominant contribution to ZFS. However, for systems containing transition metal elements, the SOC contribution to the ZFS tensor can be greater than the spin-spin contribution. The authors should make it clear that the ZFS tensor reported in this work does not represent the actual ZFS of the system, unless an argument can be made on why SOC contribution is insignificant for the systems under study.

REVIEWER COMMENTS

Reviewer #1 (Remarks to the Author):

Authors performed density functional theory (DFT) calculations to search for a new defect for qubit generation, especially at telecom wavelength. Some **qualitative (not quantitative) criteria** were selected, and because of that generalized gradient approximation calculations were able to be used to screen some of the defects at the first stage. After that hybrid DFT calculations were performed to further analyze the electronic structure quantitatively. For a few selected defects, more physical properties were investigated deeply. I have no doubt that the calculations were performed with care, however, there is **one thing that must be clarified** before the final decision.

The **formation energy of defects is too high**. For instance, the NV center has defect formation energy of more than 9 eV in the shaded region in Figure 2. The proposed W_Se defect also has high formation energy around 4 eV. This is really reasonably low value to form the defects in 2D materials? In other semiconductors, as far as I know, such defects are not considered to be formed.

Reply to reviewer #1's comment: We appreciate the positive comment on validity of our calculations. As the reviewer mentioned, the calculated formation energies of the defects are high, and in fact, such high defect formation energies are required for quantum defect applications so that the defects are not easily formed by uncontrolled environmental effects during the solid host material processing. This requirement of high defect formation energy is satisfied by the NV center ($N_C V_C^{-1}$ rather than $N_C + V_C$) with the formation energy of 4-6 eV which is comparable to that of the 2D quantum defects proposed in this work. We note that the reviewer interpreted $N_C + V_C$ in Figure 2 as the NV center defect, but the 9 eV formation energy is actually for the creation of two independent defects of N_C and V_C rather than the NV center defect itself. The energy difference of 3-5 eV between the formation energies of NV center (4-6 eV) and $N_C + V_C$ defects (9 eV) represent the binding energy of two independent defects (N_C , V_C) into $N_C V_C^{-1}$ defect complex, confirming the stability of the NV center against dissociation into separate point defects. Furthermore, the NV center formation energy of 4-6 eV has been confirmed in many other works agreeing with our calculation (Fig. 2a). As the reviewer noted, the defects with high formation energy would not form spontaneously under normal conditions, such as diamond exposed to an air environment since N_2 in air would have large thermodynamic energy barrier to be incorporated into diamond as N_C defect. Such high defect formation energy enables researchers to introduce N atoms into diamond host material by controlled ion-implantation and subsequent thermal annealing to form NV center defects. Based on these defect energetics of separate point defects and the defect complex, the diamond NV center has been a well-regarded prototype quantum defect, used for diverse quantum technologies, such as quantum metrologies. In this manuscript, we have demonstrated in Fig. 2c that the formation energy of a M_X defect (~5 eV) is lower than the sum of formation energies (~8 eV) of the two element defects of M_I and V_X ; therefore, the M_X defects will be created by annealing a system with

the controlled introduction of M_I and V_X defects. Since the existence of an antisite defect (e.g., Mo_S in MoS_2) has been demonstrated experimentally [Hong et al., Nat. Commun. 6, 6293 (2015); Khan et al., Nanophotonics 7, 1589 (2018)], M_X in MX_2 is expected to be formed with an impurity metal atom M. When it comes to defect positioning, a high formation energy is even advantageous because a high formation energy can lead to optically isolated quantum defects as we readily observe very low defect densities for the NV center [Racke et al., Appl. Phys. Lett. 118, 204003 (2021)]. As explained above, ~4 eV of the formation energy of the proposed M_X quantum defects is a reasonable value which can enable a controlled experimental creation of the quantum defects. We have added a description of the experimental relevance of the 2D quantum defects in the revised manuscript.

Modification in p. 9

Before: N/A

After: The formation of an antisite defect Mo_S in a MoS_2 , which is among the M_X defect family, has been confirmed experimentally [Hong et al., Nat. Commun. 6, 6293 (2015); Khan et al., Nanophotonics 7, 1589 (2018)]. Along with the experimental observation of Mo_S , the similar formation energy diagrams for the M_X defects in the family (Figure S3) support the feasible creation of the M_X defect family.

Reviewer #2 (Remarks to the Author):

Y. Lee and the coauthors reported first-principles density functional calculations on spin defects in TMDC materials to theoretically search for suitable spin qubit candidates, which could be optically addressable. The authors found that substitutional metal impurity for X in MX₂ TMDC hosts would produce spin-triplet ground state along with spin-conserving optical transition energy in the telecom range. Hybrid density functional calculations were employed to predict the basic single-electron band structure and the defect formation energy of the suggested defects. The authors also computed spin Hamiltonian parameters such as zero-field splitting and hyperfine parameters, and radiative recombination rates using DFT.

The **motivation of this work is interesting and timely**. A search of spin qubits in a 2D materials host is an important research topic and it gains a large amount of attention as correctly pointed out by the authors. However, I found that this **theory-only work does not support enough their main claim that the MX defect is a spin qubit candidate in the TMDC materials**. Furthermore, I found several **other critical issues in the manuscript and the work** does not meet the expected standards for high-profile journals like nature communications. Therefore, I do not recommend the publication of this work at Nature Communications. In the following, I summarize some **critical issues to be addressed**. After making the following changes and downgrading some of their strong claims, the authors should submit it to a physics journal, which better suits the overall scope of this work.

1. There are **no results and discussions on singlet shelving states**, which play a crucial role in optical initialization and readout of NV-like spin qubit systems. The authors should compute the full many-electron energy levels of the suggested defects and show compelling theoretical evidence of intersystem crossing which could lead to optical initialization in order to fully support their main claim.

Reply to reviewer #2's major comment 1: The reviewer is correct that intersystem crossings can be very important in the operation of a spin qubit. For room temperature sensing operations, this feature can enable optical initialization (at rather low fidelity) and readout (after integrating many measurement cycles). However, it is well understood that ISC are not required for low temperature initialization and readout for quantum computation and network schemes; instead, resonant, spin selective excitation is required. To appreciate this point, initialization and readout are regularly performed in trapped ion qubit systems! To this point, inter-system crossings should be avoided/minimized for high-fidelity readout in which cycling resonant transitions are required [Robledo et al., Nature 477, 574 (2011)]. A cycling transition coupled with a spin-manipulation mechanism (either via microwaves or an optical lambda system) are all that is required.

However, given the importance of ISC crossing for room temperature sensing and because they can limit read-out fidelity in low temperature computing applications, we now verify that the shelving states exist and include an estimate of the energy positions. In order to confirm the singlet shelving

states, we estimated energy positions of 1E and 1A_1 for an antisite Mo_S defect in MoS_2 using the hybrid functional without SOC. While the energy difference between 3A_2 and 3E is 1.11 eV, the computed energy difference between 3A_2 and 1E is 0.33 eV. Since the singlet state 1A_1 cannot be directly accessible with the single-particle picture in DFT, we adopted the group theoretic approach [Maze et al., New J. Phys. 13, 025025 (2011)], which demonstrates the ratio of energy difference between 1A_1 and 1E relative to that between 3A_2 and 1E is 1:2, resulting in 0.66 eV of the energy difference between 3A_2 and 1A_1 . In summary, the antisite defect has the two singlet shelving states (1E , 1A_1) between the triplet ground state (3A_2) and the triplet excited state (3E). The M_X defect family are exhibiting the similar electronic configurations to the antisite defect. Having considered that the M_X defect family has the same C_{3v} symmetry as the NV center in diamond, the triplet-singlet intersystem crossing is expected to be symmetrically allowed for our proposed M_X defects as in the NV center.

Modification in p. 17

Before: However, the intersystem crossing (ISC) transition between a triplet state and a singlet state could play a major role in a nonradiative process.

After: The intersystem crossing (ISC) transition between a triplet state and a singlet state can play an important role in a nonradiative process and can enable the low-fidelity room-temperature optical initialization and readout of the qubit-based sensors. The M_X defect family symmetrically resemble to the NV center in diamond is expected to exhibit symmetry-allowed ISCs as in the NV center [Tsai et al., Nat. Commun. 13, 492 (2022)]. We note, however, that for the high-fidelity initialization and readout required for quantum computation and network schemes, resonant, spin selective excitation is required along with avoided or minimized ISCs [Robledo et al., Nature 477, 574 (2011)].

2. Calculation and estimation of the **non-radiative lifetime** would be highly desirable to examine the possibility of the intersystem crossing. Without considering **non-radiative intersystem crossing rates**, it is impossible to imagine that this system can function as NV-like optically addressable spin qubits.

Reply to reviewer #2's major comment 2: The non-radiative intersystem crossing is an important factor for determining sensing operation, and our proposed defects resembling the NV center are expected to exhibit symmetrically allowed intersystem crossing. However, as we addressed in the previous comment, ISC is not necessarily required for spin-defect-based quantum technologies (e.g., high-fidelity quantum computation and network). Moreover, quantitative estimation of the non-radiative lifetime using the first-principles calculation remains challenging and is currently beyond the community's capability, where a proposed first-principles approach significantly overestimates a non-radiative ISC transition rate in the NV center [A. Gali, Nanophotonics 8, (2019)]. Having symmetrically allowed ISC in mind, the computation of the non-radiative intersystem crossing rates

can remain as a future work while waiting for experimental efforts for direct measurements or further advances in computational techniques.

3. The suggested defects have **very large ZFS** parameters (~10 GHz), so it is **not practically feasible** to realize spin qubits by using these defects because of experimental constraints.

Reply to reviewer #2's major comment 3: 10 GHz, while larger than the 2.9 GHz ZFS of the NV center, is well-within the experimental accessible range for spin qubits. With spin-1/2 splittings of 28 GHz/T, silicon-based spin qubits typically operate at > 10 GHz [Xue et al., Nature 601, 343 (2022)]. While more challenging than single-digit GHz operation, the requirement for reduced thermal noise of gate-defined/controlled qubits often makes these high-GHz operation a necessity. Superconducting qubits operate in the 1-12 GHz regime [Oliver and Welander, MRS Bulletin 38, 816 (2013)]. So while 10 GHz is less convenient, it is by no means impractical. Further, a higher zero-field splitting could enable high-temperature resonant spin readout as well as higher Purcell factors (coherent broadening of the optical transition via integration with a cavity for higher photon collection efficiency) while still maintaining spin selectivity.

Modification in p. 14

Before: D of the MX defect family are 10–20 GHz, about an order of magnitude larger than that of the NV center, which is helpful in carrying out spin-selective optical excitation.

After: D of the MX defect family are 10–20 GHz, about an order of magnitude larger than that of the NV center, which is within the experimentally accessible range of microwave control [Oliver and Welander, MRS Bulletin 38, 816 (2013); Xue et al., Nature 601, 343 (2022)] and could enable higher-temperature resonant spin readout as well as the compatibility of higher Purcell factors [Puecell, Phys. Rev. Lett. 69, 681 (1946)] with resonant optical spin selectivity.

4. The authors should present a detailed comparison between the **zero-phonon line and the ionization energy of the defects**. The authors computed the charge transition levels, so it is possible to compute the ionization energy by considering the bandgap of the defects. This is required to support their claim that the defects would not be ionized during optical excitation.

Reply to reviewer #2's major comment 4: We thank the reviewer for pointing this out. According to the reviewer's comment, the optical transition energy should be compared to the ionization energy of defect, not the difference of the KS eigenvalues, due to an ambiguity of the interpretation of the Kohn-Sham eigenvalue. The original description would be fine for rough estimation, but to be more precise, the ZPL energy (Table 1) needs to be compared with the ionization energy, relating to the charge transition level between the positive and neutral charge states (+/0 CTL) that can be extracted from the formation energy diagram (Figure 2). Here, the ionization energy corresponds to the energy

difference between $+/0$ CTL and CBM. Comparing the calculated ZPL and the ionization energy, it is found that the ZPL energies (e.g., 0.79 eV for W_{Se}) are smaller than the ionization energies (1.2 eV for W_{Se}), prohibiting the single-photon ionization of the defect. In order for readers to clarify the reviewer's comment, we have corrected a sentence in the revised manuscript.

Modification in p. 6

Before: The optical transitions lie within the bandgap E_g , prohibiting single-photon ionization of the defect [Figure 1(d-f)].

After: The optical transitions lie within the bandgap E_g , prohibiting single-photon ionization of the defect [Figure 1(d-f)]; since its estimation based on Kohn-Sham eigenvalues can be erroneous owing to the ambiguous interpretation of the Kohn-Sham eigenvalues, we further confirmed this from the comparison of the zero-phonon line energy and the ionization energy determined by the charge transition level, more precisely (e.g., the zero phonon line energy of W_{Se} in $MoSe_2$ is 0.79 eV, and the ionization energy of that is 1.2 eV).

5. The ground-state of **CBVN in h-BN** is not spin-triplet but spin-singlet. Please consider the latest literature on the h-BN quantum defects and take it into account.

Reply to reviewer #2's major comment 5: We greatly thank the reviewer for this comment that the ground state of CBVN in hBN could be the spin-singlet state. We now include a review article and a relevant paper mentioning the spin-singlet ground state of the defect, and have cited them in the revised manuscript:

- Reimers, J. R., Sajid, A., Kobayashi, R. & Ford, M. J. Understanding and Calibrating Density-Functional-Theory Calculations Describing the Energy and Spectroscopy of Defect Sites in Hexagonal Boron Nitride. *J. Chem. Theory Comput.* 14, 1602–1613 (2018).
- Sajid, A., Ford, M. J. & Reimers, J. R. Single-photon emitters in hexagonal boron nitride: a review of progress. *Rep. Prog. Phys.* 83, 044501 (2020),

Also, to take it account, we have made a correction to the part of the CBVN in hBN in the revised manuscript as follows.

Modification in 6

Before: In addition to the MX defects in monolayer TMDs, Table I includes our simulation results of the NV center in diamond and the CBVN defect in monolayer hBN, which have been reported to meet the aforementioned criteria [18,19,28].

After: In addition to the MX defects in monolayer TMDs, Table I includes our simulation results of the NV center in diamond [28] and the CBVN defect in monolayer hBN [18,19], which have been reported to meet the aforementioned criteria, although sophisticated approaches beyond the hybrid functional demonstrated that the ground state of the CBVN in hBN could be spin-singlet [the references above].

6. The authors claim that the MX defects could be readily created because their defect formation energy is smaller than that of the diamond NV center or that of CBVN in h-BN, and it is also lower than the sum of formation energies of the MI and VX. But, **these are not enough to support their claim**. First of all, defect formation mechanism and kinetics in h-BN ($E_g \sim 6$ eV) or in diamond ($E_g \sim 5$ eV) are **completely different** from those in TMDC materials ($E_g \sim 1 \sim 2$ eV). So the mere comparison between the defect formation energies is not enough to claim the experimental feasibility of creating MX defects in TMDC materials. Second, the authors should compare the defect formation energy of the MX defects to that of **other well-known defects or possible competing defects**. For example, would MI defects be stable enough, or was it experimentally observed? How is the defect formation energy of MX compared to that of M on the metal site?

Reply to reviewer #2's major comment 6: We agree with the reviewer that different band gaps reflect different atomic bondings of TMDC which are weaker than the diamond and hBN. For this reason, the ion-implantation and thermal annealing methods may not be directly applicable to form M_X defects in 2D MX_2 semiconductors as the reviewer noted. The defect formation mechanism of M_X defects in TMDC would be distinguished from the NV center or the hBN defect in terms of kinetics. As an effort to resolve the critical concern regarding the feasibility of the M_X defect formation, we have found experimental evidence of the presence of M_X defects in TMDC. Hong et al., Nat. Commun. 6, 6293 (2015) and Khan et al., Nanophotonics 7, 1589 (2018) have demonstrated experimentally an antisite defect in a TMDC, more specifically Mo_S in MoS_2 (one of the M_X defects). As shown in Fig. S4, the formation energy diagrams for the M_X defects in the family exhibit a similar form with 3-5 eV formation energy, which suggests that the aforementioned evidence of the existence of the Mo_S defect can be an evidence for the feasible creation of the identified M_X defects. We have added these regarding experimental evidence of Mo_S in the revised manuscript.

As the reviewer commented, there would be many different defects that are likely to form in TMDC under energetic ion irradiation conditions, and the ion implantation method used for NV center defects in diamond may in the end not be suitable to form M_X defects in 2D MX_2 semiconductors. The ability to form defects via implantation will need to be experimentally confirmed. However, the 2D nature of the materials in this work opens up a second defect formation path that is not accessible to diamond. Rather than following the ion-implantation approach, we are envisioning an atomically

precise formation of M_X defects based on controlled positioning of chalcogen vacancy (V_X) defects and subsequent metal impurity atom placement at the V_X defect sites using scanning probe microscopy [Hosaka, et. al. "Fabrication of nanostructures using scanning probe microscopes" J. Vac. Sci. Tech. B 13 (6), 2813 (1995); P. Liu et al. "First-Principle Prediction on STM Tip Manipulation of Ti Adatom on Two-Dimensional Monolayer YBr₃," SCANNING, 5434935 (2019).] Experimental evidence of Mo_S antisite defect and V_S in MoS_2 suggest possible synthesis pathways of introducing metal impurity atoms on MX_2 surface with V_X defects leading to M_X defect formation. Our recent modeling work on Ti adatom manipulation on YBr_3 surface leading to controlled positioning of the Ti atom at atomic pore site illustrates the feasibility of this alternative M_X defect formation method. This approach of controlled M_X defect formation is a topic of our current modeling and experimental studies for future publications.

Modification in p. 9

Before: N/A

After: The formation of an antisite defect Mo_S in a MoS_2 , which is among the MX defect family, has been confirmed experimentally [Hong et al., Nat. Commun. 6, 6293 (2015); Khan et al., Nanophotonics 7, 1589 (2018)]. Along with the experimental observation of Mo_S , the similar formation energy diagrams for the MX defects in the family (Figure S4) support the feasible creation of the MX defect family.

7. It seems that the authors used the HSE06 functional for all the MX_2 host materials considered in this study without adjusting the **mixing parameter and screening parameter**. Is this choice good enough to accurately predict the band-gap of all materials, which is an important quantity particularly for this study?

Reply to reviewer #2's major comment 7: As the reviewer pointed out, the mixing parameter and screening parameter affect the band gap in some degree. Nevertheless, according to a review paper dealing with comprehensive first-principles point defect calculations [Freysoldt et al., Rev. Mod. Phys 86, 253 (2014)], the HSE06 functional can be the best overall choice if the band gaps are important, and the authors of the review paper recommend using the HSE06 functional as it is (the mixing parameter $\alpha = 1/4$, the screening parameter $\omega = 0.2$). Following this suggestion, for the overall search for M_X defects in 2D MX_2 semiconductors, we adopted the HSE06 functional along with the standard parameters ($\omega = 0.2$, $\alpha = 1/4$) in order to keep our approach the first principles without using any empirical parameters. Although we could adjust the mixing parameter and screening parameter to match the band gaps of known TMDCs, it is uncertain that a defect energy level is more accurate when we adopt those parameters adjusted for band gap, and such analysis would require detailed evaluations for specific defect systems. Therefore, it is also reasonable to choose those standard parameters for the current work, in which many other properties,

such as geometry, the relative position of defect energy levels, magnetic properties, optical properties, and so forth, are of importance, as well as the band gap. As our future research further progresses with a specific defect system on a TMDC, we expect to follow reviewer's suggestion to examine the quantitative details on HSE parameters.

The followings are other minor issues.

1. In table 1, the authors reported the band gaps. Are these **theoretical band gaps**? What are the numbers in parentheses? Please describe them in detail.

Reply to reviewer #2's minor comment 1: The band gaps presented in Table 1 are theoretical values, and the numbers between parentheses are corrected by considering the spin-orbit coupling effect. As the reviewer suggested, we have described them in detail in the revised manuscript.

Modification in the caption of Table 1

Before: Defect levels between parentheses correspond to results with SOC.

After: All the values in the table were theoretically estimated in this work, and the numbers between parentheses correspond to results with SOC.

2. Please describe how the **chemical potential range for the atomic elements** was computed when computing the defect formation energies. What competing phases were considered for W, Mo, etc?

Reply to reviewer #2's minor comment 2: To determine the chemical potential range, we considered competing phases given in phase stability diagrams (see the figure below) provided by the Materials Project [APL Materials, 2013, 1(1), 011002.]. However, the values in the phase stability diagrams are given in typical DFT, not HSE hybrid functional DFT. Therefore, the quantities used for plotting the formation energy diagrams in our work were calculated by ourselves using HSE functional. According to the reviewer's comment, we have added the way of determining the chemical potential range in the revised manuscript.

Figure S2. Phase diagrams provided by Materials Project [APL Materials, 2013, 1(1), 011002.] to determine chemical potentials for the formation energy diagrams. (a) Phase diagram of a binary compound C-N. Phase stability diagrams of ternary compounds (b) B-N-C in the B-N chemical potential space, (c) Mo-S-W in the Mo-S chemical potential space, (d) W-S-Mo in the W-S chemical potential space, (e) Mo-Se-W in the Mo-Se chemical potential space, and (f) W-Se-Mo in the W-Se chemical potential space. Red dots indicate the N-rich condition and the host metal rich conditions, showing lower defect formation energies.

Modification in p. 9

Before: N/A.

After: The chemical potential range was determined by considering competing phases (Figure S2) given in phase stability diagrams provided by Materials Project [APL Materials, 2013, 1(1), 011002.]; based on the phase stability diagrams, we further computed the chemical potentials within the HSE06 hybrid functional to plot the formation energy diagrams at extreme conditions, such as the M-rich condition. The chemical potentials of C and N are obtained in the diamond crystal and the N₂ molecule, respectively

3. On page 6, the authors stated that "The optical transitions lie within the bandgap ΔE_g , prohibiting single-photon ionization of the defect [Figure 1(d-f)].", which is not true. The optical transition energy should be compared to the **ionization energy of the defect**.

Reply to reviewer #2's minor comment 3: We thank the reviewer for pointing this out. This comment is closely related to the reviewer's major comment 4. We have addressed this issue in the response to the major comment 4.

4. On page 16, the authors stated "The WSe in MoSe₂ exhibits a 4.2 μ s decay time. Overall, ΔE_R of the MX defect family is 100-1000 times larger compared with the NV center in diamond and CBVN in hBN. While (slightly) shorter ΔE_R may be desirable, we note ΔE_R is already 5 orders of magnitude shorter than the current most promising defect telecom qubit, Er³⁺:Y₂SiO₅". But I think this is not a fair comparison. The authors claimed that the MX defects would be spin qubit candidates similar to the NV center in diamond. Then, it looks strange to compare its optical property to **Er³⁺:Y₂SiO₅, which operates in a completely different way**.

Reply to reviewer #2's minor comment 4: Our goal in this paper is to demonstrate a class of promising defects for quantum information in 2D materials. A defect's promise will depend both on the targeted application as well as the combined defect properties. NV centers have relatively short optical lifetimes, small DW factor, long spin coherence and emission in visible light. Er³⁺ has very long optical lifetimes, large DW factor, long spin coherence and emits at telecom wavelength. It is precisely because of the different structures, that different properties are obtained. Here, we compare our M_X defect qubit to the rare earth qubit because it is the most mature qubit candidate that emits in the telecom-wavelength range. We do not claim that the electronic structure between the two is the same. Whereas rare earth quantum defects, such as Er³⁺:Y₂SiO₅, operate using an intra-f-shell transition, transition metal quantum defects are based on d-orbital transition. In addition, the rare earth qubits would not be affected by the materials host whereas the transition metal qubits are sensitive to the host properties. As mentioned in a review paper [Wolfowicz et al., Nat. Rev. Mater. 6, 906 (2021)], however, they still rely on the same operation principle where spin selective optical

transition plays a critical role in qubit readout and manipulation operation. In the revised manuscript, we have added a description of the electronic structure to let readers know the difference between the rare earth ion qubit and the transition metal ion qubit.

Modification in p. 16

Before: While (slightly) shorter τ_R may be desirable, we note τ_R is already 5 orders of magnitude shorter than the current most promising defect telecom qubit, $\text{Er}^{3+}:\text{Y}_2\text{SiO}_5$.

After: While (slightly) shorter τ_R may be desirable, we note τ_R is already 5 orders of magnitude shorter than the current most promising defect telecom qubit, $\text{Er}^{3+}:\text{Y}_2\text{SiO}_5$ where the intra f-shell transitions are utilized unlike the transition metal defects with d-orbital physics.

5. On page 16, the authors stated "With moderate \square_R and a

large ZFS, it is possible to achieve this enhancement while still retaining frequency-selective spin excitation for spin-photon entanglement and spin read-out." But, **there is no support for this claim**. So, this sentence should be removed.

Reply to reviewer #2's minor comment 5: According to the reviewer's comment, we provided reasonings of this claim. Support for this claim is derived from the physics of Purcell enhancement for any radiative transition. A radiative Purcell enhancement of N will result, deterministically in a broadening of the optical of N (due to the finite time-bandwidth product or the uncertainty principle). Since the zero-field splitting is large, we can indeed increase the radiative recombination rate by several orders of magnitude while still having spectrally-resolved optical transitions corresponding to the different spin states.

We have altered the sentence to make it more clear that spin-selectivity will still be retained in the presence of a large Purcell enhancement.

Modification in p. 17

Before: ... by 4 orders of magnitude. With moderate τ_R and a large ZFS, it is possible to achieve this enhancement while still retaining frequency-selective spin excitation for spin-photon entanglement and spin read-out.

After: ... by 4 orders of magnitude via the Purcell effect. Due to the large ZFS, the system should still retain frequency-selective spin excitation for spin-photon entanglement and spin read-out even with the 4 orders of magnitude frequency broadening.

6. On page 16, the authors stated "Together with the radiative process, nonradiative recombination is a vital process determining quantum yield. The absence of crossing between the potential energy curves of 3E and 3A2 shown in Figure 3(c) indicates that the nonradiative transition between the triplet states is unlikely to occur." **This is also not right. The nonradiative transition between the triplet states depends on many critical factors.** Without considering them, the authors should not claim like this.

Reply to reviewer #2's minor comment 6: We appreciate letting us know the the claim could be imprecise. We agree the reviewer's comment that the nonradiative transition depends on many other factors. Based on the reviewer's comment, we revised the claim with mentioning the caveat.

Modification in p. 16

Before: The absence of crossing between the potential energy curves of 3E and 3A2 shown in Figure 3(c) indicates that the nonradiative transition between the triplet states is unlikely to occur.

After: The absence of crossing between the potential energy curves of 3E and 3A2 shown in Figure 3(c) indicates that the nonradiative transition between the triplet states is less likely to occur; however, a further investigation is necessary to make sure the rare nonradiative transition because the transition could depend on many critical factors.

Reviewer #3 (Remarks to the Author):

In this manuscript, Lee et al. present a first-principles study of a novel family of spin-defects in TMDs. The authors reported various thermodynamic, electronic, optical, and magnetic properties of these defects obtained from DFT calculations. Overall, I think this work is sound and interesting from a first-principles perspective, but I am **not convinced of the experimental relevance of the defects proposed by this work**. Therefore, I think this work is more suitable for a more specialized journal.

My major comments are:

1. The defect formation energy reported in Fig. 2c seems to be relatively high for the new defect. I am not sure if **it is feasible to create the defects experimentally**. Without experimental evidence or convincing theoretical arguments, the existence of the defects proposed by the paper is only hypothetical.

Reply to reviewer #3's comment 1: We thank the reviewer for the positive comment on the manuscript. We have made an effort to resolve the critical concern regarding the feasibility of the defect formation despite the high formation energy. First of all, we have found experimental evidence of the pertaining defects. Hong et al., Nat. Commun. 6, 6293 (2015) and Khan et al., Nanophotonics 7, 1589 (2018) have demonstrated experimentally an antisite defect in a TMD, more specifically Mo_S in MoS₂ (one of the M_X defects). As shown in Fig. S4, the formation energy diagrams for the M_X defects in the family exhibit a similar form in 3-5 eV range, which means that the aforementioned evidence of the existence of the Mo_S defect can be grounds for the feasible creation of all the M_X defects. We have added the searched references regarding experimental evidence of Mo_S in the revised manuscript. Next, the high formation energy of the target defect complex is not an issue by itself. The diamond NV center has been a well-regarded quantum defect, used for quantum technologies, such as quantum metrologies. Even NV center exhibits 4-6 eV of formation energy, comparable to that of the quantum defects proposed here. The formation energy range 4-6 eV has been confirmed in many other works, as well as our calculation. If the formation energy of the defect complex is lower than that of element defects (e.g., N_C and V_C for NV center), sufficient annealing time and temperature end up with the formation of the desirable defect complex. In the manuscript, we have demonstrated that the formation energy of a M_X defect is lower than the sum of formation energies of the two element defects of M_I and V_X; therefore, the M_X defects will be created by annealing a system with preexisting M_I and V_X defects. When it comes to defect positioning, a high formation energy is even advantageous because a high formation energy can lead to optically isolated quantum defects as we readily observe very low defect densities for the NV center [Racke et al., Appl. Phys. Lett. 118, 204003 (2021)]. As demonstrated above, ~4

eV of the formation energy of the proposed quantum defects is a reasonable value that can bring about the formation of the quantum defect. With that in mind, the current work will possess a substantial impact on quantum defect society by proposing a feasible 2D quantum defect family, and we believe that our manuscript is suitable for publication in Nature Communications.

Modification in p. 10

Before: N/A

After: The formation of an antisite defect MoS in a MoS₂, which is among the MX defect family, has been confirmed experimentally [Hong et al., Nat. Commun. 6, 6293 (2015); Khan et al., Nanophotonics 7, 1589 (2018)]. Along with the experimental observation of MoS, the similar formation energy diagrams for the MX defects in the family (Figure S4) support the feasible creation of the MX defect family.

2. From Table 1, it seems including the SOC effect significantly shifts the DFT results for energy levels, but the calculation of **ZPL does not include the SOC effect**. It is unclear whether the new defects would still be predicted to operate on telecom wavelengths if the SOC effect is taken into account.

Reply to reviewer #3's comment 2: We appreciate pointing this out. It could be better to clarify the SOC effects on ZPL in the main text. Since the most prominent effect of SOC is the shift in the defect energy levels, the SOC effects could be approximately captured from the shift in the energy levels. Having done that, we have updated Table 1 and TOC by including the SOC-corrected ZPL and added a corresponding description of the SOC effects. As a result, the ZPL energies slightly decrease by a few tens of meV, which was only a minor shift when we consider the operation on telecom wavelengths.

Modification in p. 12

Before: The ZPL energies of the MX defect family typically lie around 1 eV, close or in the telecom band, with the calculated WSe ZPL energy at 0.79 eV.

After: In Table 1, the SOC-corrected ZPL energies between parentheses are approximated by estimating shifts in the defect energy levels shown in the same table. The ZPL energies of the MX defect family typically lie around 1 eV, close or in the telecom band, with the SOC-corrected W_{Se} ZPL energy at 0.74 eV.

3. The authors claim that "The ZFS tensor determines the dipolar spin-spin interaction between electrons". **This is not correct.** In fact, the spin-spin interaction is only one contribution to the ZFS tensor. For main group systems like diamond or hBN, the spin-spin interaction is usually the dominant contribution to ZFS. However, for systems **containing transition metal elements**, the **SOC contribution to the ZFS tensor can be greater than the spin-spin contribution.** The authors should make it clear that the ZFS tensor reported in this work does not represent the actual ZFS of the system, unless an argument can be made on why SOC contribution is insignificant for the systems under study.

Reply to reviewer #3's comment 3: We appreciate the reviewer's comment. As pointed out by the reviewer, both the spin-spin interaction and SOC determine the ZFS; thus the ZFS could be larger than the current result when a heavy element is included [Biktagirov et al., Phys. Rev. Res. 2, 023071 (2020)]. According to the reviewer's suggestion, we have made it clear that the actual ZFS could be greater than what we estimated. Even though the ZFS presented here is not the exact value, it still gives us useful information about the lower bound. We thank the reviewer again for letting us know about this omission.

Modification in p. 15

Before: N/A

After: because of the additional contribution of SOC [Biktagirov et al., Phys. Rev. Res. 2, 023071 (2020)], the ZFS could be even greater than the value presented in Table 1 especially with a heavy element, such as W.

REVIEWER COMMENTS

Reviewer #1 (Remarks to the Author):

Three reviewers including me mentioned the high formation energy of defects, so this is probably the weakest point of the manuscript. I think that the authors' response is somewhat reasonable, so I would like to recommend this manuscript for publication in Nature Communications.

Reviewer #2 (Remarks to the Author):

Y. Lee and the coauthors revised the manuscript based on the questions and suggestions that I provided. However, I found that most of the key issues that I raised were not addressed, which I describe in detail below. Therefore, I do not recommend its publication in Nature Communications.

1. For a potential qubit initialization and readout scheme, the authors seem to propose to use cycling transitions based on resonant spin selective excitations, which are used in trapped ion qubit systems. First of all, the suggested hypothetical defect candidate is not a trapped ion system, but a defect system embedded in a solid-state matrix. Therefore, to show that this initialization and readout scheme would work, the authors should provide quantitative analysis and evidence to support their claim, which is not found in the manuscript.

2. In the response, the authors estimated the position of the singlet states by using the group theoretical approach developed by Maze et al. However, this method used several approximations to construct an effective Hilbert space for the NV center in diamond and to describe its electronic structure, which would not be simply transferable to defects in 2D materials. So, I am not convinced by this argument proposed by the authors.

3. There are many factors to be considered for understanding intersystem crossing processes. Symmetry is one. But, what about the transition rates involved in the process? These transition rates strongly depend on the electronic structure of a defect and play key roles in the intersystem crossing process. Without any calculations or supporting evidence, I am not convinced by the authors' argument and speculation.

4. The ground-state of the CBVN in h-BN is known to be spin-singlet and one can get this state in DFT by just putting two electrons in the a1 state in Fig. 1(e). This state is just a closed-shell spin-singlet state and doesn't require any other sophisticated quantum chemistry method.

5. I also suggested comparing the defect formation energy of the suggested defect candidate to those of other competing defects, which was not done in the response. Instead, the author suggested looking at other defect systems in MoS₂ and YBr₃. So, I am not convinced that this defect candidate could be created in an experiment and I think the issue remains unresolved.

Overall, I am NOT persuaded that this manuscript warrants its publication in Nature communications.

Reviewer #3 (Remarks to the Author):

I have no further concerns about the technical aspects of the first principles calculations performed in this work.

I would be more confident to recommend this manuscript to be published in more specialized journals such as NPJ Computational Materials or other NPJ journals. The editor and other referees can decide whether the defects reported in this manuscript is of enough experimental relevance to be appealing to the broad audience of Nature Communications.

Reply to the reviewers' comments on NCOMMS-21-21535A by Yeonghun Lee et al.

Reviewer #1 (Remarks to the Author):

Three reviewers including me mentioned the high formation energy of defects, so this is probably the weakest point of the manuscript. I think that the authors' response is somewhat reasonable, so I would like to recommend this manuscript for publication in Nature Communications.

Reply to reviewer #1's comment: We thank the reviewer for recommending the publication of the manuscript.

Reviewer #2 (Remarks to the Author):

Y. Lee and the coauthors revised the manuscript based on the questions and suggestions that I provided. However, I found that most of the key issues that I raised were not addressed, which I describe in detail below. Therefore, I do not recommend its publication in Nature Communications.

1. For a potential qubit initialization and readout scheme, **the authors seem to propose to use cycling transitions based on resonant spin selective excitations, which are used in trapped ion qubit systems.** First of all, the suggested hypothetical defect candidate is not a trapped ion system, but a defect system embedded in a solid-state matrix. Therefore, **to show that this initialization and readout scheme would work, the authors should provide quantitative analysis and evidence to support their claim, which is not found in the manuscript.**

Reply to reviewer #2's comment 1: We have seriously taken into account the reviewer's suggestion and have performed quantitative analysis of the intersystem crossing process to demonstrate the potential for room temperature initialization and readout. The calculation was combined with the quantum chemistry approach to deal with the many-body state, and the computation process has been adopted in [Tyler J. Smart, Kejun Li, Junqing Xu, and Yuan Ping, npj Comput. Mater. 7, 59 (2021)]. Our approach reproduces reasonably well the experimental ISC transition rate of the NV center in diamond (3E to 1A_1 : 30.6 MHz vs. 60.7 MHz [J-P Tetienne et al 2012 New J. Phys. 14 103033]). We find that the simulated transition rate of ISC from the spin-triplet excited state to the spin-singlet shelving state is 32 MHz = $(0.031 \mu\text{s})^{-1}$ in W_{se} , which is larger than the inverse radiative

lifetime $0.24 \text{ MHz} = (4.2 \text{ } \mu\text{s})^{-1}$ of the spin-triplet excited state. The large ratio of the ISC:radiative rates indicates that the proposed quantum defect can exhibit the initialization and readout operation via the spin-selective decay pathways.

We respect the reviewer's comment on the cycling transitions based on resonant spin selective excitations. The trapped ion is a leading platform utilizing the resonant spin selective excitations for initialization and readout. The resonant spin selective excitations have also been widely adopted for single-shot detection of defect qubits in solid-state matrix [Neumann et al, Science 329, 542 (2010); Robledo et al., Nature 477, 574 (2011); Pfaff et al., Nat. Phys. 9, 29 (2013)]. Since we do not directly demonstrate such high-fidelity operation, we remains it as an example of the case where the ISC should be suppressed for the $m_s=0$ transition which is not theoretically allowed in the ideal, unstrained case..

Modification in p.19 of the main text

Before: The intersystem crossing (ISC) transition between a triplet state and a singlet state can play an important role in a nonradiative process and can enable the low-fidelity room-temperature optical initialization and readout of the qubit-based sensors. The MX defect family symmetrically resemble to the NV center in diamond is expected to exhibit symmetry-allowed ISCs as in the NV center⁷².

After: ISC is mediated by a combination of spin-orbital coupling (SOC) and electron-phonon interaction. The crossing rate was calculated by the application of Fermi's golden rule according to the formula^{75,83}:

$$\Gamma_{\text{ISC}} = 4\pi\lambda_{\perp}^2 \tilde{X}_{if},$$

$$\tilde{X}_{if} = \sum_m w_m \sum_n |\langle \phi_{im} | \phi_{fn} \rangle|^2 \delta(\Delta E_{if} + m\hbar\omega_i - n\hbar\omega_f),$$

Where λ_{\perp} is the transverse SOC constant between spin single and spin triplet states, X_{if} is the phonon wavefunction overlap between initial state i with phonon quantum number m and final state f with phonon quantum number n , ϕ_{im} and ϕ_{fn} are the phonon wavefunctions, ω_i and ω_f are the phonon frequencies, w_m is the occupation number of phonon according to Bose-Einstein distribution, and ΔE_{if} is the energy difference between the initial state and final state (See Methods for further details of phonon wavefunction overlap and SOC strength calculations). The ISC from the triplet excite states 3E to the singlet shelving state 1A1 can be symmetrically allowed when $m_s=\pm 1$ ⁷². The simulated transition rate of ISC from the triplet excited state to the singlet shelving state is $32 \text{ MHz} = (0.031 \text{ } \mu\text{s})^{-1}$ in WSe in MoSe₂, which is larger than the inverse radiative lifetime $0.24 \text{ MHz} = (4.2 \text{ } \mu\text{s})^{-1}$ of the triplet excited state. The large ratio of the ISC:radiative rates is a pre-

requisite for initialization and readout operation via the spin-selective decay pathways (Figure 4). We note, however, that for the high-fidelity initialization and readout required for computation and network, resonant, spin selective excitation is required along with avoided or minimized ISC⁷⁵ for at least one of the spin states (e.g., $m_s=0$). Since SOC underlies the ISC transition⁷³, we will be able to engineer ISC by utilizing various transition metal dopants with different SOCs.

Figure 4. Sublevel structure of WSe in MoSe2. The radiative processes are shown in the orange line. The blue dashed lines show the symmetry-allowed ISC transitions from the triplet excited state 3E to the singlet state 1A_1 and the transition from 1A_1 to 3A_2 , which are responsible for spin-selective decay, enabling the initialization and readout operations. The purple circular arrows within ZFS indicate the manipulation of qubit states by microwave.

Modification in the method section

Before: N/A

After:

Phonon wavefunction overlap and SOC strength. ISC is attributed to a combination of SOC and electron-phonon interaction. To obtain the phonon wavefunction overlap between initial and final state, one-dimensional harmonic oscillation approximation was used which introduces the general configuration coordinate diagram. The potential surfaces of spin-triplet excited state 3E and spin-singlet state 1A_1 were obtained by linearly interpolating between initial 3E and final 1A_1 structures involved in the ISC. Energies of interpolated structure were calculated using constrained-occupation DFT⁷⁴. Since Kohn-Sham DFT theory cannot describe states composed of multiple Slater determinates, approximate electron occupations— $|\uparrow a_1 e_x \rangle$ for 3E and $|\uparrow e_x e_y \rangle$ for 1A_1 —were adopted, and we made an approximation to access the energy of the 1A_1 at the equilibrium geometry following Mackoit-Sinkeviciene et al.⁸³ All constrained DFT computations were performed

using VASP, facilitated by modified Nonrand84 preprocessing and postprocessing for interpolated structure energy calculation. The calculated configuration coordinate diagram for 3E and 1A1 is shown in Figure S7S7.

SOC strength was computed with the ORCA code⁸⁵ using time-dependent density functional theory (TDDFT)⁸⁶. Different from VASP, ORCA does not have the feature of periodic boundary condition. We thus constructed cluster models for both NCVC-1 and WSe defects by cutting relaxed structures from bulk and saturating dangling bonds to reproduce the electronic structures of bulk structures. The dangling bonds in diamond cluster are easily saturated by H while TMD is well known of complicated edge states and charge transfer between edges and defects for over 10 Å⁸⁷. After testing with different size, boundary, and termination groups, a cluster with hybrid zigzag and arm-chair boundary and termination groups of H, OH and NH was found using B3LYP functional to have both the same spin density as periodic result [Figure S8S8(a, b)] and HOMO-LUMO gap of 1.22eV to get reasonable excited states [Figure S8S8(c)]. We obtained SOC values of 4.71 GHz for λ_{\parallel} and 44.6 GHz for λ_{\perp} for NCVC-1 defect using PBE functionals with def2-TZVP basis, which agrees well with previously computed values and experimentally measured values^{73,74,88}. With the calculated λ_{\perp} , we obtained the 3E \rightarrow 1A1 ISC rate for NV center in diamond at 30.6 MHz which is in a fair agreement with literature reported value of 60.7 MHz⁸⁹. We then computed the SOC strength for the axial λ_{\parallel} and non-axial λ_{\perp} components of the WSe defect in MoSe₂ using B3LYP functionals to be 69 and 109 GHz, respectively.

Figure S7. Configuration coordinate diagrams for 3E and 1A1 state of WSe defect in MoSe₂.

Figure S8. Structures and spin densities for (a) periodic and (b) cluster WSe defect in MoSe2 (isosurface level = 0.00296 e/bohr³). (c) Densities of states as a function of energy relative to the Fermi level for WSe cluster using B3LYP functional

2. In the response, the authors estimated the position of the singlet states by using the group theoretical approach developed by Maze et al. However, this method used several approximations to **construct an effective Hilbert space for the NV center in diamond** and to describe its electronic structure, which **would not be simply transferable to defects in 2D materials**. So, I am not convinced by this argument proposed by the authors.

Reply to reviewer #2's comment 2: Our 2D quantum defects show the same symmetry C_{3v} with their NV center. Looking at the KS energy levels and their occupation in Figure 1, the hole configuration of the 2D defect is identical to the electron configuration of the NV center. Since hole and electron representations are totally equivalent, the group theoretical approach should be transferrable. In the approach, however the effect of the other electrons is not considered, but a calculation based on many-body perturbation theory support the conclusion on the ordering of singlet states [Ma et al., Phys. Rev. B 81, 041204(R) (2010)]. We further verify the singlet shenving states using another method introduced in [Mackoit-Sinkevičienė et al., Appl. Phys. Lett. 115, 212101 (2019)]. Constrined-occupation DFT was used for the energy calculation of spin-singlet states. The electron configuration for 1E is $^1E: |e_x \bar{e}_x\rangle$ where e_x and \bar{e}_x indicate the two spin channels of e_x orbital, and the electron configuration for 1A_1 states is $^1A_1: \frac{1}{\sqrt{2}}(|e_x \bar{e}_y\rangle + |e_y \bar{e}_x\rangle)$. 1E is single Slater determinant state and can be calculated by Kohn-Sham DFT theory, whereas 1A_1 state is composed of multiple Slater determinants and therefore cannot be treated with DFT calculation. Here we made an approximation to access the energy of 1A_1 following Mackoit-Sinkevičienė et al. The three components of the ground spin-triplet state are $|T;0\rangle = \frac{1}{\sqrt{2}}(|e_x \bar{e}_y\rangle - |e_y \bar{e}_x\rangle)$, $|T;-1\rangle = |\bar{e}_x \bar{e}_y\rangle$,

$|T;1\rangle = |e_x e_y\rangle$. A simple math derivation leads to $E(^1A_1) = E\left(\frac{1}{\sqrt{2}}(|e_x \bar{e}_y\rangle + |e_y \bar{e}_x\rangle)\right) = 2E(|e_x \bar{e}_y\rangle) - E(|T;0\rangle) = 2E(|e_x \bar{e}_y\rangle) - E(T)$. Since the spin-triplet state energy is known, and $|e_x \bar{e}_y\rangle$ is a single Slater determinant which can be described by DFT, the energy of 1A_1 can be calculated. The calculated energy levels of the singlet shelving states 1E and 1A_1 are respectively 0.240 eV and 0.475 eV with respect to the triplet ground state 3A_2 , verifying the singlet shelving states between the triplet ground state 3A_2 and the triplet excited state 3E .

3. There are many factors to be considered for understanding intersystem crossing processes. Symmetry is one. But, what about the **transition rates involved in the process**? These transition rates strongly depend on the electronic structure of a defect and play key roles in the intersystem crossing process. Without any calculations **or supporting evidence**, I am not convinced by the authors' argument and speculation.

Reply to reviewer #2's comment 3: We thank the reviewer for the valuable suggestion, which indeed has improved our paper. We have performed the intersystem crossing transition rate, which is crucial to show that the initialization and readout scheme would work in the proposed defects. Please refer to the reply to the comment 1 for detailed discussion.

4. The ground-state of the CBVN in h-BN is known to be spin-singlet and **one can get this state in DFT by just putting two electrons in the a1 state in Fig. 1(e)**. This state is just a closed-shell spin-singlet state and doesn't require any other sophisticated quantum chemistry method.

Reply to reviewer #2's comment 4: As the reviewer mentioned, a closed-shell spin-singlet state would have shown the spin singlet as the ground state. However, the hybrid functional HSE06 within VASP did not provide spin-singlet ground state, shown in the table below adapted from [Reimers et al., J. Chem. Theory Comput. 14, 1602–1613 (2018)]. The error here is from the multireference nature of the spin-singlet ground state. Quantum chemistry approaches can treat significant multireference character and showed that the spin-singlet state could be the ground state. To better appreciate it, we revised our manuscript.

Table 1. Energies (in eV) of Various States of the $V_N C_B$ Model Compound (Figure 1b) with Respect to $(1)^1A_1$ Evaluated at the Test Geometry^a

state	MRCI ^b	CASPT2 ^c	CCSD	CCSD (T)	EOM-CCSD-S	EOM-CCSD-T	HSE06 VASP	HSE06 G09	HSE06 TD-S	HSE06 TD-T	CAM	CAM TD-S	CAM TD-T
$(2)^1A_1$		2.30	2.73	2.03			2.06	2.11			2.61		
$(1)^1B_1$	1.45	0.94			1.25			0.63	0.62		0.89	0.91	
$(2)^1B_1$		2.98			3.44			2.57	2.61		3.05	3.06	
$(3)^1B_1$		4.24			4.65				3.62			4.28	
$(4)^1B_1$		4.68							4.96			5.53	
$(1)^1B_2$	5.19	4.22			4.79			3.89	3.96		4.36	4.56	
$(2)^1B_2$	5.15	4.37			4.62			3.79	3.84		4.34	4.40	
$(1)^1A_2$	3.28	3.03			3.36			2.34	2.22		2.74	3.01	
$(2)^1A_2$					4.36				3.37			4.02	
$(1)^3A_1$	3.96	3.50	3.97	3.66		4.01	3.09	3.17		3.14	3.63		3.62
$(2)^3A_1$									4.17	4.25		4.87	4.92
$(3)^3A_1$									4.37	4.49		5.16	5.23
$(4)^3A_1$									4.69	4.81		5.19	5.45
$(5)^3A_1$		5.22	5.57	5.15		5.07		4.75		5.05	5.30		5.34
$(1)^3B_1$	0.32	0.02	0.24	0.24	0.25	[0.24]	-0.20	-0.16	-0.59	[-0.16]	0.09	-0.61	[0.09]
$(2)^3B_1$		2.34			2.63	3.01	1.99	2.07	1.80	2.10	2.52	2.20	2.62
$(3)^3B_1$		4.33							3.49	3.57		4.16	4.16
$(4)^3B_1$										4.49		5.00	5.04
$(5)^3B_1$		5.22							4.49			5.26	5.33
$(1)^3B_2$			4.19	4.01	4.21	4.59		3.55	3.35	3.57	3.91	3.75	4.21
$(2)^3B_2$	4.96	4.66	4.74	4.42	4.60			3.91	3.76	3.95	4.46	4.32	4.97
$(1)^3A_2$	3.27	2.42	3.01	2.85	3.32	3.18	2.23	2.29	2.15	2.26	2.70	2.94	2.83
$(2)^3A_2$					4.33	4.29			3.29	3.38		3.94	3.89
$(3)^3A_2$		3.96	5.25	4.65				4.05		3.96	4.78		4.76

^aSee SI for geometry used. CAM = CAM-B3LYP; EOM-CCSD and TDDFT calculations are performed from both the $(1)^1A_1$ reference state ("S") and the $(1)^3B_1$ reference state ("T"). ^bAverage of five calculations using different active spaces, ± 0.07 eV. ^cAverage of two calculations using different active spaces, ± 0.3 eV.

Modification in p. 7

Before: although sophisticated approaches beyond the hybrid functional demonstrated that the ground state of the CBVN in hBN could be spin-singlet [26,40].

After: although quantum chemistry approaches beyond the hybrid functional demonstrated that the ground state of the CBVN in hBN could be spin-singlet by taking into account multireference nature of the singlet state [26,40].

5. I also suggested comparing the defect formation energy of the suggested defect candidate to those of other competing defects, which was not done in the response. Instead, the author suggested looking at other defect systems in MoS2 and YBr3. So, I am not convinced that this defect candidate could be created in an experiment and I think the issue remains unresolved.

Reply to reviewer #2's comment 4: In the current revision, we have performed defect formation energies calculations of possible competing defects and compared the defect formation energy of the suggested defect candidate to those of other competing defects, following the reviewer's suggestion. In the host material, MoSe₂, the target quantum defect is W_{Se}, which can be formed

from a combination of V_{Se} and W_I . The additional calculation results of the possible competing defects are shown in the figure below. It is not surprising that the formation energy of the W_{Se} is higher than W on the metal site, W_{Mo} , because the electron configuration of W is similar to that of Mo . However, V_{Se} exhibits a lower defect formation energy, making it much easier to be formed than V_{Mo} . Furthermore, the formation energy of W_I is comparable to that of Mo_I . Thus, once we introduce W_I in the presence of abundant V_{Mo} , the W_{Se} defect complex will be readily formed.

Modification in p. 11

Before: V/A

After: Figure S5 shows defect formation energies of possible competing defects, where V_{Se} is much easier to be formed than V_{Mo} ; thus, once we introduce W_I in the presence of abundant V_{Mo} , the W_{Se} complex can be readily formed

The figure below has been added in SI.

Figure S5. Defect formation energies of other competing defects with WSe in MoSe_2 under the host's Mo-rich condition. V_{Se} is much easier to be formed than V_{Mo} , and the stability of W_i is almost the same as that of Mo_i . Thus, once we introduce W_i in the presence of abundant V_{Mo} , the WSe complex can be readily formed.

Overall, I am NOT persuaded that this manuscript warrants its publication in Nature communications.

Reviewer #3 (Remarks to the Author):

I have no further concerns about the technical aspects of the first principles calculations performed in this work.

I would be more confident to recommend this manuscript to be published in more specialized journals such as NPJ Computational Materials or other NPJ journals. The editor and other referees can decide whether the defects reported in this manuscript is of enough experimental relevance to be appealing to the broad audience of Nature Communications.

Reply to reviewer #3's comment: We thank the reviewer for confirming the technical soundness of the manuscript.

REVIEWERS' COMMENTS

Reviewer #2 (Remarks to the Author):

The authors partially addressed some of the issues that I have raised in terms of the potential qubit initialization mechanism and defect formation energy. The other important issues that I have raised were not addressed, such as the result related to the CBVN defect.

Most importantly, however, I found that a paper was recently published in Nature Communications, which reported basically the same result:

Antisite defect qubits in monolayer transition metal dichalcogenides

J. Tsai et al., Nat. comm. 13, 492 (2022).

This paper is also cited in the resubmitted manuscript as Ref. 72. In this paper, J. Tsai and the co-authors already predicted the same metal antisite defect in TMDC (transition metal ion substituting for an X ion in MX₂ materials) as qubit candidates. They also reported many key results, which substantially overlap with the content of the resubmitted manuscript, such as defect level diagrams, defect formation energies, charge transition levels, ZPLs, and analysis of intersystem crossing processes.

Therefore, many key results of the resubmitted manuscript are the reproduction of the previous results reported by J. Tsai et al, which is unfortunate. I conclude that the resubmitted manuscript lacks a novelty, which is required for its publication in Nat. Comm. I recommend transferring this manuscript to more technical journals such as Communications Physics.